# Emerging green steel markets surrounding the EU emissions trading system and carbon border adjustment mechanism

Constantin Johnson [ORCID], Max Åhman [ORCID] ✉, Lars J. Nilsson & Zhenxi Li [ORCID]

The global steel industry accounts for 8–10 % of global $CO_2$ emissions and requires deep decarbonisation for achieving the targets set in the Paris Agreement. However, no low-emission primary steel production technology has yet been commercially feasible or deployed. Through analysing revisions and additions of European Union climate policy, we show that green hydrogen-based steelmaking in competitive locations achieves cost-competitiveness on the European market starting 2026. If the deployment of competitive low-emission steelmaking is insufficient, we show that the European steel industry loses competitiveness vis-à-vis countries with access to low-cost renewable energy. Therefore, we assess the options for the European steel industry to relocate the energy-intensive ironmaking step and trade Hot Briquetted Iron for rapid deep decarbonisation of the European steel industry. Lastly, we discuss complementing policy options to enhance the Carbon Border Adjustment Mechanism's strategic value through European Union-lead global climate cooperation and the possibility of sparking an international decarbonisation race.

There has been a consistent expansion of climate mitigation policies and laws in the world since the Paris Agreement, although ambition and progress vary across countries and sectors[1]. The steel industry accounts for 8–10 % of global $CO_2$ emissions[2] and 5 % within the European Union[3] (EU). As one of the most widely traded commodities globally, the competitiveness of steel and the decarbonisation of its production are strongly influenced by international trade dynamics and carbon policies. It is in this evolving context that the EU develops and adapts its climate policies, most recently manifested through the introduction of a Carbon Border Adjustment Mechanism (CBAM) to prevent carbon leakage while maintaining high ambitions for reducing emissions. The steel industry represents the largest share of CBAM goods, accounting for 69 % by reported weight[4]. In this paper, we deploy a dynamic optimisation model to assess the cost-competitiveness of green steel on global markets created by the revised EU Emissions Trading System (ETS) and the CBAM.

There is a limited set of technological options to reduce emissions for steelmaking by 85 % or more. Secondary steel production using scrap in an electric arc furnace (EAF) offers lower emissions, but is constrained by the availability and quality of scrap to replace iron ore-based steel (primary steel). Therefore, around 50 % of all steel production in year 2050 is projected to come from iron ore[5,6]. For primary steel, the main options include carbon capture and storage (CCS) on plants that use fossil coal or natural gas, and hydrogen-direct reduced iron ($H_2$-DRI) with green hydrogen as a reducing agent. While past CCS projects have frequently failed[7], emission capture rates are also constrained by technical and spatial limitations[8–10]. Therefore, the $H_2$-DRI route has emerged as the most promising option[11] due to increased cost-competitiveness and higher emission reduction potential. In our analysis we therefore assume that $H_2$-DRI-EAF is the only low-emission alternative to conventional primary steelmaking in the period 2025–2035. In this study, we provide an assessment of how recent EU ETS revisions and the CBAM impact the cost-competitiveness of $H_2$-DRI-EAF steelmaking on the global market. While natural gas (NG) is often viewed as a transitional option, a full assessment of transitional

Division of Environmental and Energy Systems Studies, Lund university, Lund, Sweden. ✉e-mail: max.ahman@miljo.lth.se

decarbonisation pathways and their associated challenges is beyond the scope of this paper.

Shifting from coal-based steel to steel based on renewable electricity changes the comparative advantages of steelmaking between countries. This has spawned a growing literature on relocation and reconfiguration of the iron and steel value chain[12–14]. Access to low-cost renewable electricity or hydrogen is an endowment that could give key regions comparative advantages for producing green iron and steel[15,16]. Access to scrap is also a future comparative advantage[5].

The idea of carbon border tariffs, taxes, or adjustment mechanisms to compensate for large differences in carbon pricing across jurisdictions, goes back to the early 1990s, when carbon pricing was increasingly seen as the way to internalise the social cost of carbon. The EU ETS has seen a price between $€60 − 100\ t_{CO_2}^{-1}$ for the past five years. However, to shield EU industries from what has been described as unfair competition[17] and avoid carbon leakage, the EU has compensated trade-exposed industries, such as the steel industry, by allocating the emission allowances (EUAs) for free. As the temperature targets of the Paris Agreement demand zero emissions by year 2050, an EU ETS combined with free allocation will no longer be an effective option to drive transformation[18]. To achieve drastic emission reductions without risking carbon leakage, the EU has decided to phase out the free allocation and phase in the CBAM concurrently from year 2026 to 2034 to level the playing field[19,20].

The competitiveness of H₂-DRI-EAF has been explored using bottom-up optimisation models with[21–23] and without[15,24–26] international trade; however, without considering the CBAM and details of the EU ETS. The potential effects of the EU CBAM have been studied by use of general equilibrium models[27,28] and through socio-economic and political perspectives[29–31]. On steel specifically, the CBAM has been studied with an input-output model[32] and a price-floor model

specifically investigating the cost-competitiveness of H₂-DR-based steel in Germany[33]. However, these studies neither capture the final details of the EU ETS and CBAM legislation described in the Methods section, nor do they include techno-economic analysis of investment options to mitigate the rising costs associated with the EU ETS or the CBAM. A significant revision of the EU ETS is that the H₂-DR technology will receive free allocation[34] which effectively subsidises production of H₂-DR-based steel in the time-period from year 2026 to 2033. Thus, the EU ETS and CBAM together create a shift in the global competitive landscape. EU ETS product benchmark revisions have been found to improve the cash flow of a low-emission steel plant in Germany[35], but without considering external market conditions. Here, we show how steel production and investments are influenced by international competitiveness and trade, together with the comparative advantages of steelmaking across countries.

We assess the market-based cost-competitiveness of steelmaking and possible reconfigurations of value chains by using a bottom-up, techno-economic dynamic optimisation model which includes an investment model and a production system and market model. Our model solves for optimal investments, steel production, and trade of steel, scrap, and iron ore, by considering material availability, investment and production costs, and current steel assets (see Fig. 1). Besides the EU27, our model includes detailed analysis of the eight largest steel producing countries and Australia, covering over 90 % of the current global steel industry. To assess potential carbon leakage, our model scenario setting consists of two main scenarios with different CBAM emission coverages: one where the CBAM only applies to direct emissions (scope 1) and another where indirect emissions (scope 2) are included. We first present an analysis of how the revised EU ETS impacts cost-competitiveness of steelmaking in the EU before presenting model results.

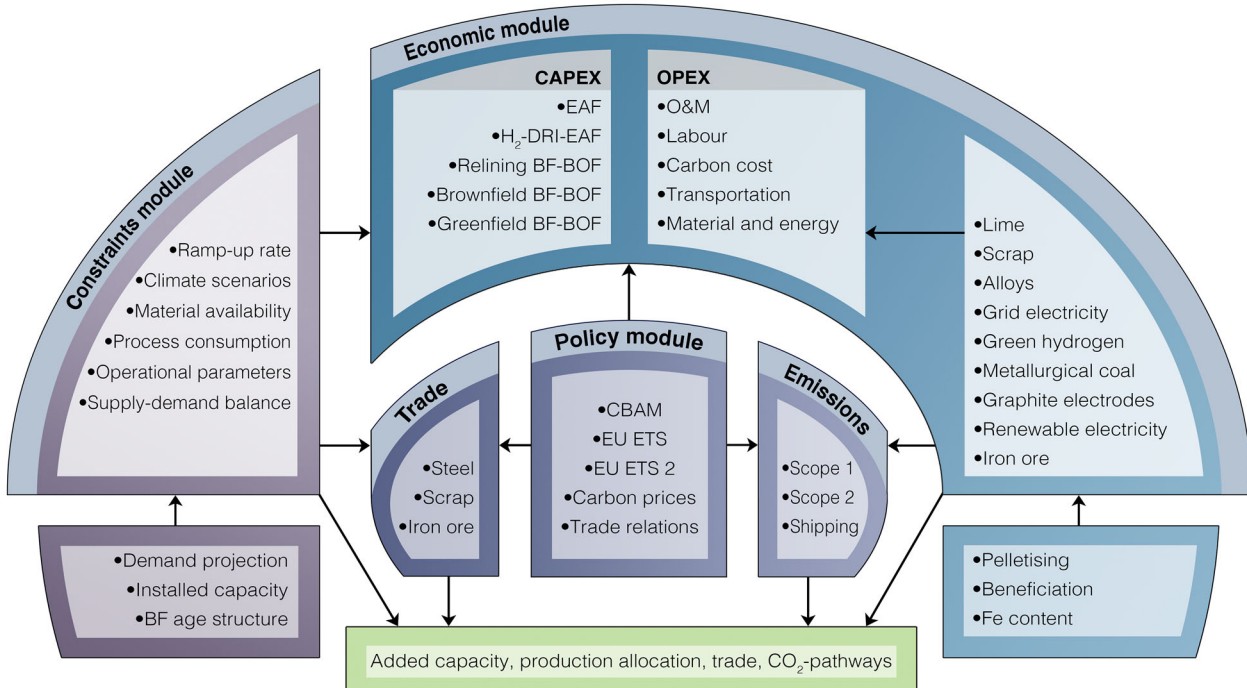

**Fig. 1 | Detailed schematic of the global steel model.** The schematic illustrates how the model combines policy, economics, trade, and resource constraints to quantify how the European Union Emissions Trading System (EU ETS) and the Carbon Border Adjustment Mechanism (CBAM) reshape competitiveness. The economic module contains capital expenditure (CAPEX) and operating expenditure (OPEX). CAPEX covers the steel production routes blast furnace-basic oxygen furnace (BF-BOF), scrap-electric arc furnace (scrap-EAF), and green hydrogen-direct reduced iron-electric arc furnace (H₂-DRI-EAF). OPEX includes fixed OPEX, such as operations and maintenance (O&M), and variable OPEX. For the specific assumptions, we refer to the chapters Methods and Supplementary Methods.

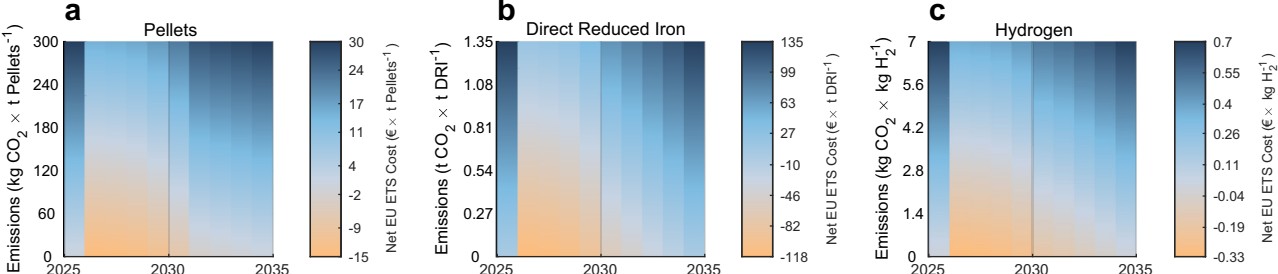

**Fig. 2 | Impact of the 2024 EU ETS revision on the net EU ETS cost for producing intermediate products for steelmaking. a–c** Values shown for a constant European Union Emissions Trading System (EU ETS) price of €100 $t_{CO_2}^{-1}$. **a** The inclusion of pelletising in the sintered ore benchmark enables free allocation for pellet production starting 2026. **b** The inclusion of direct reduced iron (DRI) reactors and sponge iron to the hot metal benchmark enables free allocation for the production of DRI starting 2026. **c** The addition of electrolysis to the hydrogen benchmark enables green hydrogen to receive free allocation starting 2026.

## Results

### Impact of EU ETS revision on steelmaking costs in the EU

The revisions of the EU ETS will effectively subsidise the production of low-emission intermediate products used in H$_2$-DRI-EAF steelmaking during 2026–2033. From 2026, the updated EU ETS benchmark definitions for sintered ore, hot metal, and hydrogen will enable free allocation of EUAs for the production of pellets, DRI, and green hydrogen, whereas emissions from these processes were fully subject to an EU ETS cost prior to 2026. As the benchmark values are based on incumbent emission-intensive production and are updated gradually to reflect lower emitting technologies (see Supplementary Methods 1: Free allocation), producers receive more EUAs for free than needed to cover emissions if the processes emit less than the benchmark value, which we refer to as a net negative EU ETS cost (see Fig. 2).

Both the conventional blast furnace-basic oxygen furnace (BF-BOF) route and the H$_2$-DRI-EAF route receive free allocation for iron based on the hot metal benchmark, which results in the highest net negative EU ETS cost for the H$_2$-DRI-EAF route if iron ore is reduced with low emissions (see Fig. 2b). Although future hot metal benchmark values are uncertain, the competitiveness of H$_2$-DRI-EAF relative to BF-BOF is invariant to the benchmark value as the net negative EU ETS cost lost from a lower benchmark value equals the additional carbon cost borne by the BF-BOF route. Corresponding logic applies to the sintered ore benchmark if the BF-BOF and H$_2$-DRI-EAF routes use equal amounts of agglomerated iron ore (sinter or pellets). Steel production through H$_2$-DRI-EAF benefits from free allocation from the hydrogen benchmark and the assumed maximum update rate for the hydrogen benchmark is hence a conservative assumption for its competitiveness, while the BF-BOF route benefits from free allocation through the coke benchmark which is not phased out.

These changes in free allocation introduce a shift in the competitive landscape of the EU market in favour of H$_2$-DRI-EAF, whereas BF-BOF and NG-DRI-EAF are disadvantaged due to EU ETS costs (see Fig. 3). Their relative competitiveness is mainly determined by the combination of electricity and natural gas prices, emissions, and EU ETS costs. For example, the hydrogen component of the synthetic gas produced from the natural gas for carburising DRI in the high-emission H$_2$-DRI-EAF scenario, replaces 11 % of the green hydrogen required for reduction in the low-emission scenario, but will not receive free allocation under the hydrogen benchmark. As a result, the production costs are lower for the high-emission scenario (€626 − 743 $t_{CS}^{-1}$) than for the low-emission scenario (€636 − 762 $t_{CS}^{-1}$), while the total costs are higher due to the higher EU ETS cost (see Fig. 3b, c).

The production costs for NG-DRI-EAF range €604 − 685 $t_{CS}^{-1}$; however, the EU ETS costs are significantly larger than for H$_2$-DRI-EAF routes. At lower EU ETS prices of €80 $t_{CO_2}^{-1}$, the low-cost and low-emission NG-DRI-EAF route achieves competitiveness with both H$_2$-DRI-EAF routes at an electricity price of €100 MWh$^{-1}$ (see Fig. 3). However, none of these routes are competitive with the BF-BOF route at this low EU ETS price. We further refine the analysis of competitiveness with the BF-BOF route in Figure 4.

The required levelised cost of hydrogen (LCOH$_2$) for competitiveness between low-emission H$_2$-DRI-EAF and BF-BOF increases by €2.12 − 3.95 $kg_{H_2}^{-1}$ from 2025 to 2026 at EU ETS prices of €80 − 150 $t_{CO_2}^{-1}$ (see Fig. 4a), reflecting the impact of the revised benchmark system boundaries. At a constant EU ETS price, the competitiveness of H$_2$-DRI-EAF declines slightly (see Fig. 4a, b) due to that the free allocation for hydrogen is phased out whereas BF-BOF benefits from free allocation for coke and lime.

Figure 3e, f show that the NG-DRI-EAF route does not achieve competitiveness with the BF-BOF route even under optimistic assumptions of low emissions and low natural gas prices of €30 MWh$^{-1}$. This is further confirmed by Fig. 4c, where the closest natural gas price to €30 MWh$^{-1}$ (€23.6 MWh$^{-1}$) only achieves competitiveness at an EU ETS price of €150 $t_{CO_2}^{-1}$ after 2030. Moreover, the NG-DRI-EAF route is noncompetitive with BF-BOF at any natural gas price if upstream emissions and methane are included in the EU ETS (see Fig. 4d). In contrast, from 2022 to 2024, average natural gas prices in Germany and France ranged €48.25−64.5 MWh$^{-1}$ and €57.65−73.5 MWh$^{-1}$ excluding taxes and levies, and €67.05−84.05 MWh$^{-1}$ and €79.55−94.8 MWh$^{-1}$ including taxes and levies[36]. We therefore conclude that the NG-DRI-EAF route will not achieve market-based competitiveness with the BF-BOF route over the period 2025−2035. Since this study investigates market-based competitiveness and our analysis shows that the NG-DRI-EAF route does not achieve marginal abatement with the BF-BOF route, the NG-DRI-EAF route has been excluded from the model, since the cost-minimising optimisation algorithm would yield no investments in the EU. For competitiveness of NG-DRI-EAF in the context of EU trade partners we refer to Supplementary Figs. 1, 2 and the Discussion chapter on EU imports with the CBAM.

### Competitiveness on the EU market

The CBAM with scopes 1 and 2 prices third-country emissions equal to the EU ETS when emission factors are equal to those of the EU, and therefore does not affect the net competitiveness—providing a level playing field. Hence, the competitive landscape and resulting trade intensity is only affected by the CBAM if third-country emissions are lower or higher than that of the EU, providing either a lower or a higher carbon cost than the EU ETS on EU production. This effect is amplified as the CBAM is gradually phased in and free allocation is phased out, making competitiveness increasingly dependent on the emission intensity of steel production. Consequently, the market advantage increasingly lies in decarbonisation to remain below the marginal market price caused by carbon costs from the EU ETS and the CBAM. Therefore, replacement of BF-BOF installations in the EU with

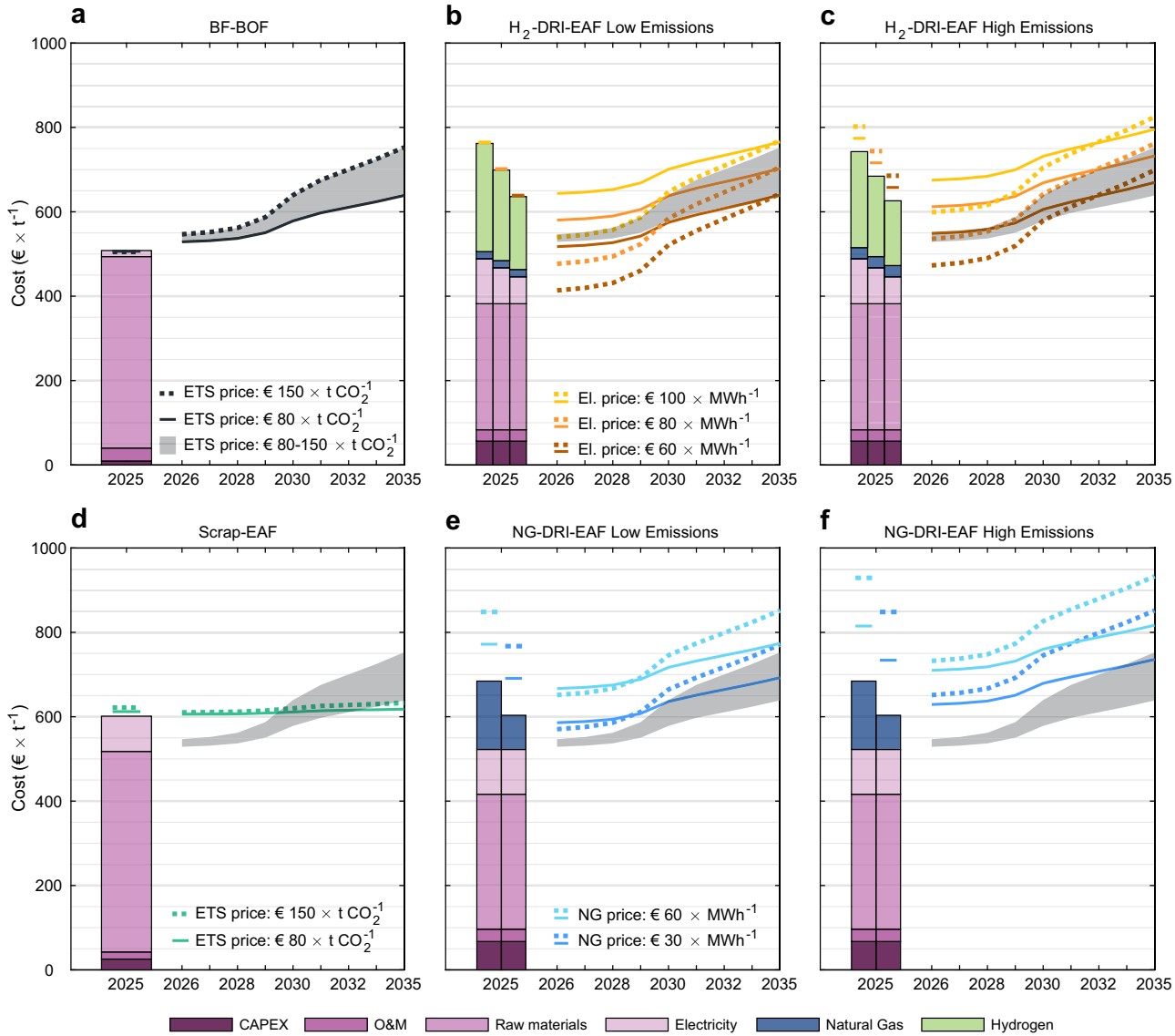

**Fig. 3 | Production costs and total costs for EU crude steel production under the phase-out of free allocation. a–f** The steel production routes shown are basic oxygen furnace-based steel (BOF), which uses pig iron produced in a blast furnace (BF) as input, and electric arc furnace-based steel (EAF), which uses either 100 % scrap or direct reduced iron (DRI) produced using green hydrogen ($H_2$) or natural gas (NG) as a reducing agent. Lines depict total cost which is production cost and net EU Emissions Trading System (ETS) cost. Solid lines are total cost with an EU ETS price of €80 $t_{CO_2}^{-1}$ and dashed lines are total cost with an EU ETS price of €150 $t_{CO_2}^{-1}$. Shaded area is total cost of BF-BOF-produced steel with an EU ETS price ranging from €80−150 $t_{CO_2}^{-1}$. Grid fees are excluded from electricity and hydrogen prices. **a–c, e, f** A scrap charge of 12.5 % scrap was assumed for these production routes.

**a** Capital expenditure for BF-BOF relining assumed to €64 $t_{CS}^{-178}$. **b, c** Three cases with electricity (El.) prices of €60, €80, and €100 MWh$^{-1}$ for EU ETS prices of €80 and €150 $t_{CO_2}^{-1}$. Both scenarios have a 1.5 % carbon content embedded in the direct reduced iron (DRI) from natural gas at a price of €60 MWh$^{-1}$. **b** Low-emission $H_2$-DRI-EAF scenario with an emission factor of 64 kg $CO_2$ $t_{CS}^{-1}$. **c** High-emission $H_2$-DRI-EAF scenario with an emission factor of 437 kg $CO_2$ $t_{CS}^{-1}$. **e, f** Two cases with natural gas prices of €30 and €60 MWh$^{-1}$ for EU ETS prices of €80 and €150 $t_{CO_2}^{-1}$. **e** Low-emission NG-DRI-EAF scenario with an emission factor of 1.13 t $CO_2$ $t_{CS}^{-1}$. **f** High-emission NG-DRI-EAF scenario with an emission factor of 1.67 t $CO_2$ $t_{CS}^{-1}$, representing an extension of the EU ETS to methane based on the 20-year global warming potential (GWP$_{20}$) of methane and upstream emissions with 3 % leakage.

competitive low-emission steel capacity is required to sustain the current primary steel production volume of 80 million tonnes per annum (Mt p.a.). However, the EU is at a disadvantage in this regard, having to decarbonise more steel capacity than each individual exporter to the EU. Additionally, third countries can collectively decarbonise more steel capacity than they currently export to the EU—gaining market share by outcompeting EU producers if sufficient EU BF-BOF installations are not replaced with competitive low-emission steel capacity. Moreover, as others also decarbonise, the margin shifts lower, requiring further emission reductions to maintain market share —this could create a dynamic of increasing decarbonisation efforts, that is, a decarbonisation race.

Europe has heterogenous conditions for renewable electricity and green hydrogen[37,38], resulting in $H_2$-DRI-EAF production costs to vary across countries and regions. From the hydrogen prices necessary for competitiveness (see Fig. 4a), we select €4.12 kg$_{H_2}^{-1}$ and €5.5 kg$_{H_2}^{-1}$ for model simulation, comparable to those attainable in northern Scandinavia, Portugal, and Spain[39–41].

Both model scenarios of CBAM emission coverage show that adding secondary steel and $H_2$-DRI-EAF capacity are the most cost optimal investment routes over the period 2026−2035. The updated EU ETS benchmark definitions push the $H_2$-DRI-EAF route into an early commercialisation phase, which indicates that the first movers to invest in $H_2$-DRI-EAF capacity in competitive locations such as northern

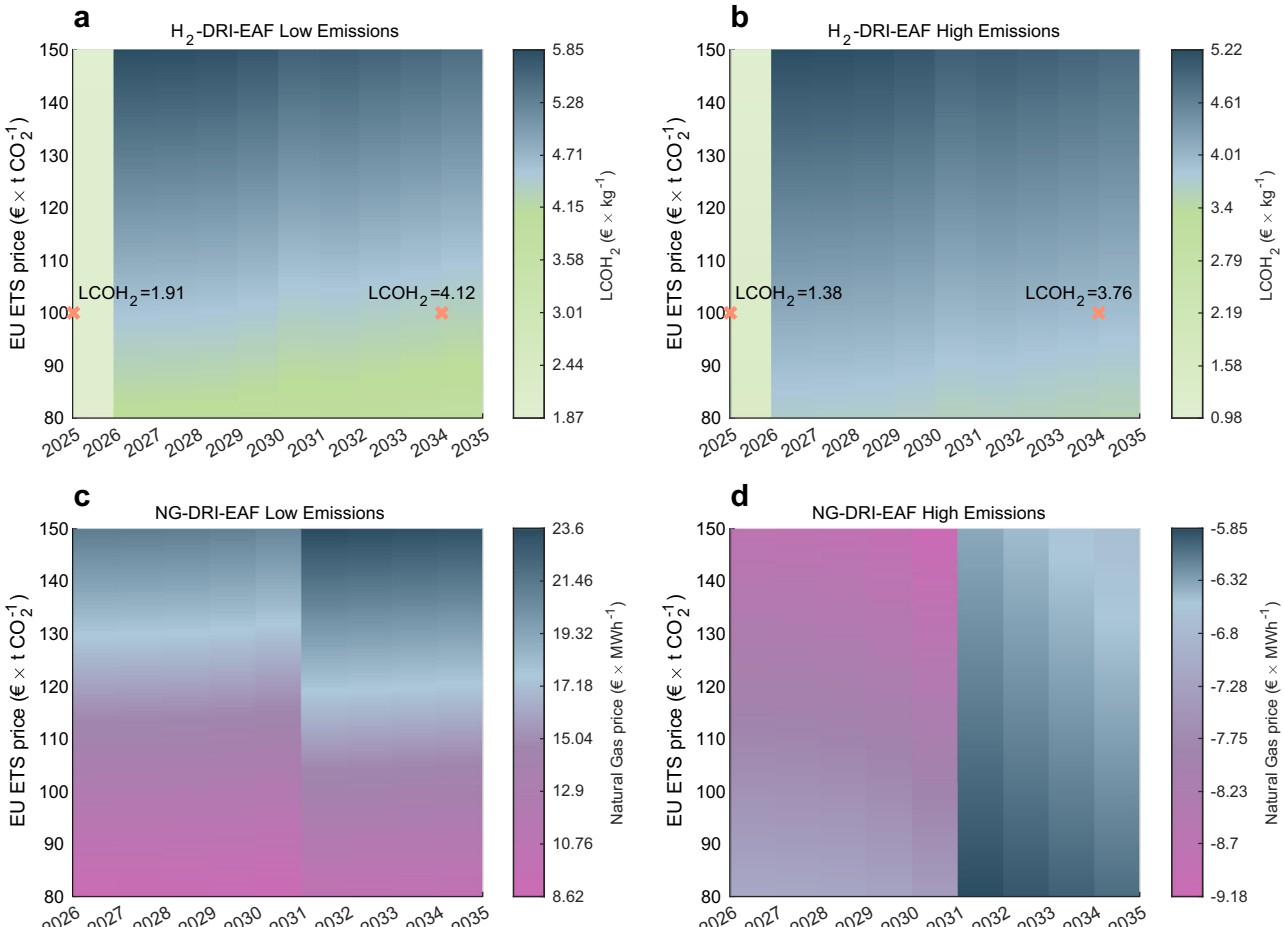

**Fig. 4 | Marginal abatement costs for H₂-DRI-EAF and NG-DRI-EAF with respect to BF-BOF in the EU.** Marginal abatement costs are shown for electric arc furnace-based steel (EAF), which uses direct reduced iron (DRI) produced using green hydrogen (H$_2$) or natural gas (NG) as a reducing agent, relative to conventional steel production through blast furnace-basic oxygen furnace (BF-BOF). European Union Emissions Trading System (EU ETS) prices range €$80-150\,t_{CO_2}^{-1}$ and examples of the levelised cost of hydrogen (LCOH$_2$) required for marginal abatement are shown for EU ETS prices of €$100\,t_{CO_2}^{-1}$. **a–d** Equal scrap charge of 12.5 % and capital expenditure for BF-BOF relining at €$64\,t_{CS}^{-1}$ were assumed. Grid fees are excluded from electricity and hydrogen prices. **a** Low-emission H$_2$-DRI-EAF scenario with an emission factor of 64 kg CO$_2$ $t_{CS}^{-1}$. **b** High-emission H$_2$-DRI-EAF scenario with an emission factor of 437 kg CO$_2$ $t_{CS}^{-1}$. **c** Low-emission NG-DRI-EAF scenario with an emission factor of 1.13 t CO$_2$ $t_{CS}^{-1}$. For 2025, the values range €$-67.22$ to €$-38.31$ MWh$^{-1}$. **d** High-emission NG-DRI-EAF scenario with an emission factor of 1.67 t CO$_2$ $t_{CS}^{-1}$. For 2025, the values range €$-97.18$ to €$-54.29$ MWh$^{-1}$.

Scandinavia, Portugal, and Spain will have market advantage in the EU (see Fig. 5). Additionally, EU's low grid-emission intensity enables cost-competitive secondary steel production, which through an estimated increased domestic scrap availability achieves a production increase from 54 Mt year 2025 to 68 Mt year 2035 (see Fig. 5). Consequently, steel produced through the BF-BOF route sets the marginal price on the EU market. The first movers to H₂-DRI-EAF capacity will likely operate with a higher scrap charge (e.g. 50 %) as scrap is cheaper than H₂-DRI in the early commercialisation phase and will in that case receive more free allocation from the EAF-benchmark, but less per tonne of steel.

The model results show that the EU market faces considerable stress post year 2029 attributed to the non-linear decline of the CBAM factor (see Fig. 6). Specifically, BF-BOF-produced steel will be non-competitive with low-emission imports year 2030 and onwards for any amount that does not receive full free allocation at an EU ETS price above €$150\,t_{CO_2}^{-1}$, and from year 2033 at an EU ETS price above €$90\,t_{CO_2}^{-1}$, thereby leading to a significant production decline (see Figs. 5, 6).

With a limited amount of competitive H₂-DRI-EAF steel available in the EU to replace BF-BOF installations, an increasing amount of imported steel will be necessary to satisfy the steel demand (see Fig. 5).

As the carbon costs from the EU ETS and the CBAM increase over time, low-emission secondary steel and H₂-DRI-EAF steel become increasingly competitive. Hence, high-emission BF-BOF imports are first substituted with lower emission BF-BOF imports, which are subsequently substituted with secondary steel imports and finally substituted with low-emission secondary steel and H₂-DRI-EAF imports. This dynamic becomes stronger with an increased EU ETS price, leading to a greater incentive for decarbonisation and therefore greater reduction in imported embedded emissions. With this incentive structure, the EU ETS and CBAM will create a de facto green materials club[42–44].

The CBAM was found to be fully effective in protecting against carbon leakage in both scenarios of CBAM implementation. However, the inclusion of scope 2 emissions further mitigates imported embedded emissions (12.9 Mt CO$_2$) due to the higher grid-emission intensity of third countries compared to the EU27 average. Moreover, the carbon cost difference by EU's lower grid-emission intensity compared to third countries only partially offsets EU's higher grid electricity prices and labour costs compared to third countries. EU secondary and H₂-DRI-EAF steel production is therefore at competitive disadvantage with a CBAM scope 1 coverage only. No change in production or trade of BF-BOF was found as a result of electricity prices

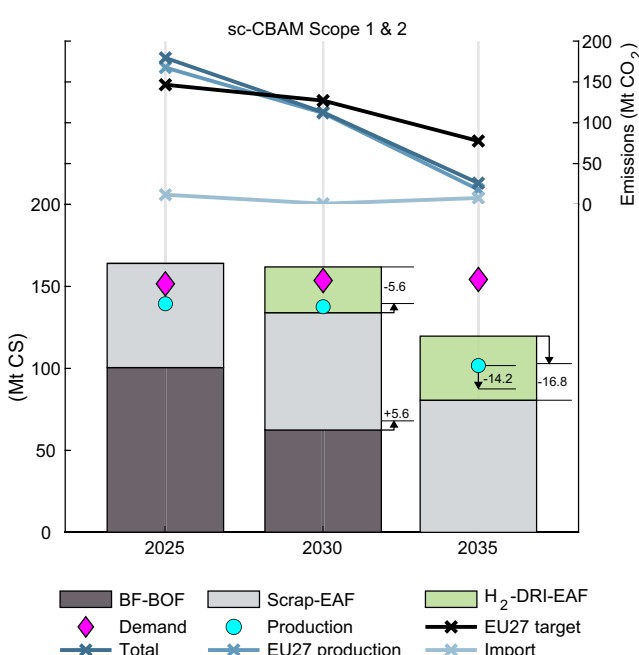

**Fig. 5 | EU27 production capacity change from year 2025 to 2035.** Model results of installed capacity, total production volume, and steel demand for EU27 with the CBAM scope 1 & 2 scenario in units of million tonnes of crude steel (Mt CS). Resulting emissions are in units of million tonnes of $CO_2$. The steel production routes shown are blast furnace-basic oxygen furnace (BF-BOF), scrap-electric arc furnace (scrap-EAF), and green hydrogen-direct reduced iron-electric arc furnace ($H_2$-DRI-EAF). The bars show installed and added steel capacity for a levelised cost of hydrogen (LCOH) input value of €4.12 $kg_{H_2}^{-1}$ and indicated changes for a LCOH of €5.5 $kg_{H_2}^{-1}$. The capacity is related to the production volume with a utilisation rate of 85 %.

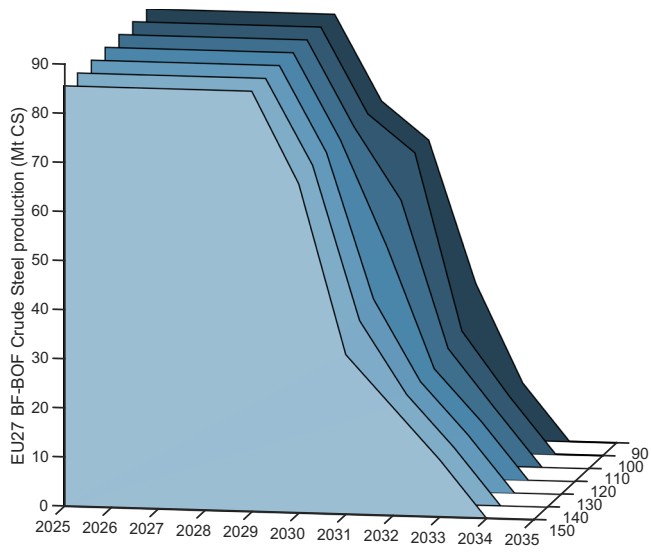

**Fig. 6 | EU27 BF-BOF steel production change.** The figure shows the model results for how the amount of competitive EU27 blast furnace-basic oxygen furnace (BF-BOF) steel production changes with the phase-out of free allocation at different European Union Emissions Trading System (EU ETS) prices (€90 − 150 $t_{CO_2}^{-1}$) for sc-CBAM scope 1 & 2. Units in million tonnes of crude steel (Mt CS).

due to its low electricity consumption (see Supplementary Table 6) and therefore also invariant to the CBAM emission coverage.

## Global green steel markets and EU CBAM impact

The global steel production system will change the coming years to 2035 as a consequence of the emergence of scrap-EAF and $H_2$-DRI-EAF as competitive low-emission options. Global variations in secondary and $H_2$-DRI-EAF steel developments are attributed to different natural endowments such as iron ore quality and potential for affordable renewable energy, as well as scrap availability. The cost-competitiveness between the BF-BOF and $H_2$-DRI-EAF routes without regard for carbon prices is mainly determined by the difference between coal and coke prices, and renewable electricity prices, since BF-BOF consumes its energy mainly from coal (and subsequently coke) and $H_2$-DRI-EAF from renewable electricity. In contrast, the cost-competitiveness of $H_2$-DRI-EAF across countries is mainly determined by electricity prices. These variations in comparative advantages, together with variations in emission intensities, change the relative advantages of competitive exports to the EU with the CBAM (see Figs. 7, 8).

We find that the CBAM has a substantial impact on the cost-competitiveness of steel exports to the EU (see Figs. 7, 8). Although BF-BOF steel produced in China has a higher emission factor than BF-BOF steel produced in the EU and thus a higher carbon cost, it is initially able to sustain cost-competitive exports to the EU market with the carbon costs from the CBAM due to its lower production cost of €461 $t_{CS}^{-1}$ compared to €508 $t_{CS}^{-1}$ of the EU (see Fig. 7a). While China's CBAM cost for BF-BOF steel is lower than the EU ETS cost for EU BF-BOF steel if scope 2 emissions are excluded from the CBAM, it becomes non-competitive on the EU market due to secondary and $H_2$-

DRI-EAF-produced steel becoming increasingly competitive options. Unlike in the case of China, India's low BF-BOF production cost of €456 $t_{CS}^{-1}$ only partially offsets the high CBAM cost attributed to its high emission factor (see Fig. 7b).

Shipping costs excluding costs from EU ETS 2 ranged €4.44−39.27, corresponding to transport distances of 991−37,635 km. Furthermore, we found that shipping costs are cheaper than iron ore beneficiation costs for Fe content differences of more than 7 % (see Supplementary Fig. 3).

Brazil, the U.S., Japan, and China were found to be the most competitive exporters to the EU of secondary steel with the CBAM (see Fig. 7), primarily due to low scrap prices. The U.S. can increase the amount of competitive export to the EU by investing in $H_2$-DRI-EAF capacity made favourable through competitive utility solar PV and onshore wind. Additionally, this potential for affordable renewable electricity enables grid decarbonisation favourable for competitive secondary steel export to the EU through its large fleet of installed EAF capacity and scrap recycling practices. While Brazil has opportunities for competitive low-emission secondary steel production, its production volume is limited by domestic scrap availability. In scenarios where Turkey's possibilities for $H_2$-DRI-EAF capacity are hindered, Turkey suffers a production decline (see Fig. 9), as the CBAM becomes stronger over time due to the declining CBAM factor. In combination with projected increases of carbon prices in Turkey which disadvantages domestic BF-BOF steel production, Turkey loses its status as net exporter unless decarbonisation efforts are undertaken, which are however, feasible due to competitive electricity from onshore wind and unexplored opportunities for offshore wind, albeit increasing the cost of hydrogen in Fig. 7g. If competitive primary steel export is sought, Turkey's low reserve of high-quality iron ore requires an investment decision of either iron ore import or hot briquetted iron (HBI) import. Critical to the largest exporter of secondary steel to the EU[45], Turkey must decarbonise its electricity sector to remain competitive with secondary steel export under the CBAM, as the possible inclusion of scope 2 emissions to the CBAM favours countries with low grid-emission intensities.

China, India, Brazil, and Australia achieved $H_2$-DRI-EAF investments without any or significant carbon prices due to their potential for affordable renewable electricity and were thus able to remain

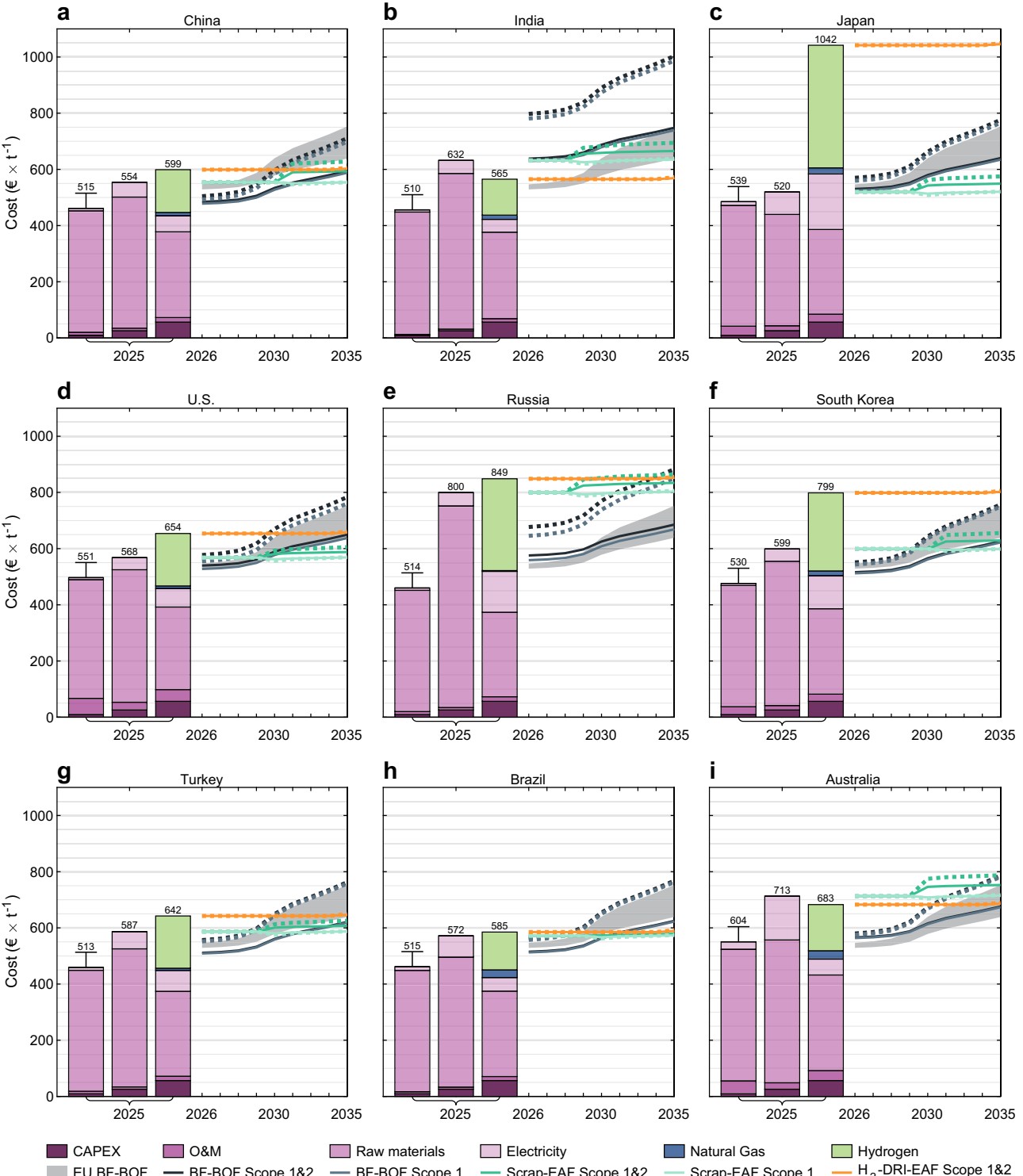

**Fig. 7 | Production costs and total costs for EU trade partners under the phase-in of the CBAM. a–i** From left to right, bars show crude steel production costs for blast furnace-basic oxygen furnace (BF-BOF), scrap-electric arc furnace (scrap-EAF), and green hydrogen-direct reduced iron-electric arc furnace (H$_2$-DRI-EAF). All costs are shown without domestic carbon prices and transportation costs. Lines depict total cost which is production cost and cost incurred from the Carbon Border Adjustment Mechanism (CBAM), excluding domestic carbon prices and transportation costs. Solid lines are total cost with a European Union Emissions Trading System (EU ETS) price of €80 t$_{CO_2}^{-1}$ and dashed lines are total cost with an EU ETS price of €150 t$_{CO_2}^{-1}$. Shaded area is total cost of EU27 BF-BOF production with an EU ETS price ranging from €80 − 150 t$_{CO_2}^{-1}$. The BF-BOF production cost shows the difference in capital investment cost between relining and greenfield investments of €53.8 t$_{CS}^{-1}$ and the total costs are shown with the capital investment cost for relining.

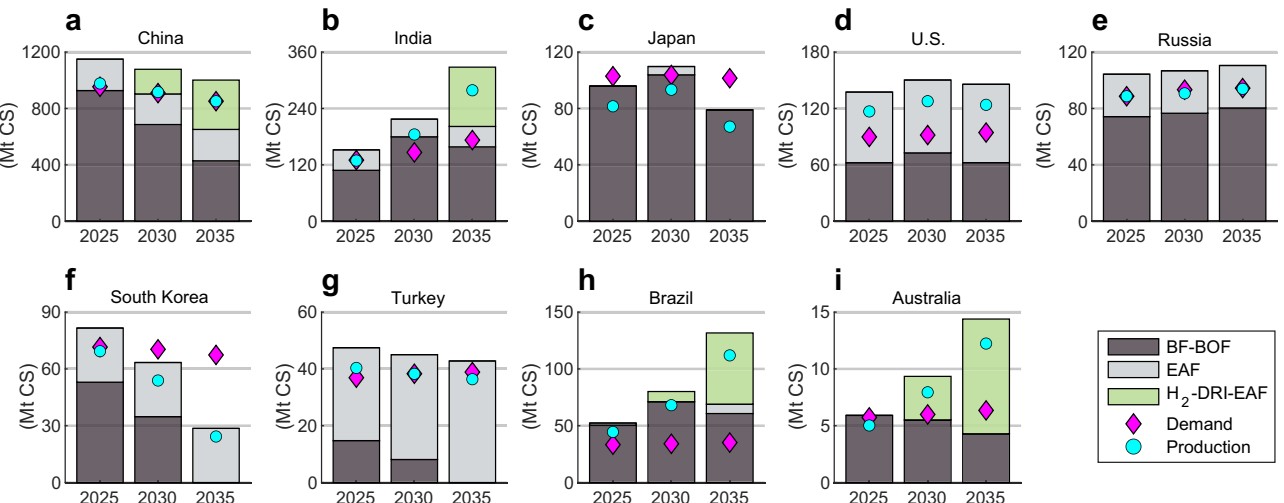

**Fig. 8 | EU27 production and import changes from year 2025 to 2035.** Import and production of crude steel through the routes blast furnace-basic oxygen furnace (BF-BOF), scrap-electric arc furnace (scrap-EAF), and green hydrogen-direct reduced iron-electric arc furnace (H₂-DRI-EAF). The model with the scenario sc-CBAM scope 1 & 2 shows that through the phase-out of free allocation and the phase-in of the Carbon Border Adjustment Mechanism (CBAM), the emission-intensive steel production and import is replaced by more competitive low-emission secondary (scrap-EAF) and H₂-DRI-EAF steel imports (renewables pull). Units in million tonnes of crude steel (Mt CS).

**Fig. 9 | Global steel production capacity from year 2025 to 2035 for the scenario sc-CBAM scope 1 & 2. a–i** The steel capacity is shown for the production routes blast furnace-basic oxygen furnace (BF-BOF), scrap-electric arc furnace (scrap-EAF), and green hydrogen-direct reduced iron-electric arc furnace (H₂-DRI-EAF). The production volume correlates to the steel capacity with an 85% utilisation rate, mothballed or excess capacity is not displayed. The comparative advantages for H₂-DRI-EAF steel production result in industrial relocation. Units in million tonnes of crude steel (Mt CS).

competitive on the EU market with H$_2$-DRI-EAF steel exports. Furthermore, a synergy between the countries able to considerably increase their H$_2$-DRI-EAF capacity and countries with domestic potential for affordable renewable electricity was identified, indicating a renewables pull effect. Although H$_2$-DRI-EAF investments were found cost-competitive in the previously mentioned countries, competitive BF-BOF steel and increasing global demand resulted in additional brownfield and greenfield BF-BOF investments in Brazil and India, further increasing existing carbon lock-ins and path-dependencies. Although India has highly competitive utility solar PV and onshore wind, access to renewable electricity at low cost for the steel industry is hindered due to a complex cross-subsidisation scheme that requires reform[46,47].

Both Japan's and South Korea's investment opportunities in H$_2$-DRI-EAF capacity were disadvantaged by high renewable electricity prices and insubstantial iron ore deposits, in conjunction with their geographical proximity to China – disadvantages ultimately attributed to geospatial factors. Japan was further disadvantaged by high grid electricity prices, seeking scrap-for-steel trade with China instead to avoid high grid electricity prices and labour costs. Carbon prices in South Korea renders steel production with the BF-BOF route non-competitive domestically, whereas secondary production is favourable through low grid electricity prices, albeit not renewable. The region is however scrap-scarce, which limits the possibility for significant secondary steel production volumes and results in a net-import pathway for South Korea. Overall, Japan and South Korea struggle to retain all production steps of primary steel production in a Paris-compatible transition unless renewable electricity or investments are heavily subsidised. However, outsourcing the energy-intensive production step of reducing iron ore to DRI to countries such as Australia, which has high-quality iron ore deposits and affordable renewable electricity, allows for a more cost-competitive global solution[12,48,49] and could be enabled through an East-Asia-Pacific green steel club.

Our model assumes perfect markets and foresight due to the optimisation algorithm which negates global overproduction with perfect pricing of steel and emissions. Additionally, by not accounting for HBI-trade in the model, the results for several countries highlight barriers for domestic H$_2$-DRI-EAF investments which could be overcome by importing HBI. In addition to being an enabler for Japan and South Korea to decarbonise their steel industries without compromising domestic steel production volume[23], HBI-trade could enable the EU to retain its current steel production volume with the phase-out of free allocation, which we elaborate on in the Discussion chapter below.

## Discussion

The EU results show favourable investments in H$_2$-DRI-EAF at hydrogen prices comparable to those that can be attained in northern Scandinavia, Portugal, and Spain[39–41], while low-emission imports are more competitive than BF-BOF and H$_2$-DRI-EAF at hydrogen prices comparable to those in e.g. Germany, France, Belgium, and Poland[39,41]. The incumbent emission-intensive steel industry in the EU will therefore likely be outcompeted by low-emission steel from third countries if the phase-out of free allocation is stronger than the phase-in of competitive H$_2$-DRI-EAF capacity. However, to align with climate targets and avoid inefficient subsidy races[50], a renewables pull can enable rapid deep decarbonisation of the European steel industry by outsourcing the energy-intensive step of ironmaking to countries and regions with quicker access to cost-competitive green hydrogen, whilst retaining the competitive aspect of steel quality and value added of downstream processing, along with the most labour intensive production[51]. Simultaneously, this serves as a key opportunity for a just transition beyond the borders of the EU.

We differentiate three different types of industrial relocation for rapid deep decarbonisation of the European steel industry. First, the European steel industry can optimise its location and reconfigure its value chains within the EU. The synergy between regions with affordable renewable electricity and regions with land areas sufficient in size for large-scale wind power and solar PV installations promotes locations such as northern Scandinavia, Portugal, and Spain. Second, HBI can be imported from established steelmaking countries with existing steelmaking competences and possibility for affordable renewable electricity. Model results highlighted China, India, the U.S., Brazil, and Australia as cost-competitive candidates while the literature also highlights Chile[15], Canada[15,52], and the Middle East and Northern Africa region[53–55]. Third, H$_2$-DRI can be outsourced by European steelmakers to developing countries with no or minor current steel production that have the opportunity for affordable renewable electricity, for example Mauretania, Namibia, Oman, Morocco, and South Africa[15,16,53–55]. Therefore, in addition to supporting rapid decarbonisation of the European steel industry, the industrial relocation could contribute to Paris-aligned sustainable development in the countries outsourced. Technology transfer and capacity building could subsequently spill over and foster energy-related climate mitigation efforts in other sectors, contributing to their Sustainable Development Goals and Nationally Determined Contributions. Optimally, countries with high-quality iron ore deposits in addition to availability of affordable renewable energy are especially favoured[15,56], and would enable co-location of the entire ironmaking step of the value chain, however not critical as the energy savings from integrated production and transportation costs for import of iron ore pellets are comparatively low. For both the second and the third option, a CBAM complemented with climate trade diplomacy, could enable critical investments that break the current carbon path-dependency and pivot the steel industry of other countries into decarbonisation and enable development through a green industrialisation.

Nykvist et al.[57] nuance the strength of a renewables pull with the role of policy to mitigate global cost differences of H$_2$-DRI-EAF steel production through a "strategic push". While we show how the design of Emission Trading Systems can provide a short-term push to enable business cases in regions with high comparative advantages in the EU, it is insufficient in mitigating large differences of H$_2$-DRI-EAF production costs. Furthermore, renewable electricity prices depend primarily on geospatial factors like solar irradiation and wind power density, and can therefore only be partially influenced by policy measures. Consequently, green hydrogen risks remaining neither competitive nor widely available in the medium or long term in high-cost industrial regions[58]. Although the main drivers influencing cost of H$_2$-DRI-EAF steel production are the comparative advantages through geospatial factors and industrial policy used as a competitive advantage, industrial policy alone is likely insufficient in enabling competitiveness in high-cost regions. For example, hydrogen prices in Germany are estimated at €8.9 − 9.2 kg$_{H_2}^{-1}$[39,41]. To achieve a hydrogen price of €4 − 4.5 kg$_{H_2}^{-1}$ with a hydrogen consumption of 50−60 kg$_{H_2}$ t$_{CS}^{-1}$, these differences correspond to €220−312 t$_{CS}^{-1}$, likely requiring substantial subsidies over the investment lifetime—exceeding the scope of a "strategic push".

Investments that are not competitive on the market could be enabled through significant state aid or trade defence instruments (e.g. tariffs) to mitigate international competition. However, this would distort competition within the European Economic Area, compete with other public spending priorities, or conflict with international principles of fair trade. With no guarantee for medium- or long-term competitiveness, sustained subsidies for production could extend over several decades and risk supporting existing fossil-based production if business cases fail to materialise. Even if multiple projects were enabled through such measures in the EU, they would likely fall short

of replacing the current primary steel production of 80 Mt p.a.—creating an iron gap.

Considerations of security and military have recently emerged as political arguments for domestic steel production[57,59], as exemplified in the communication of the Steel and Metals Action Plan where the European Commission outlines that a tank consists of 50–60 tonnes of steel[60]. With these metrics, 10,000 tanks amount to 0.5–0.6 Mt of steel which corresponds to only 0.6 % of EU primary steel production and 0.4 % of total steel production, allowing for decarbonisation through a combination of pathways, like HBI import, which preserve domestic steel production without compromising climate targets. Additionally, given the constrained availability of resources in EU Member States, substantial imports of iron ore, coal, or natural gas would remain necessary depending on the technological path chosen. With strong bilateral relationships becoming increasingly important in a geopolitically fragmented world, resilience could instead lie in diversified supply chains and bilaterial or multilateral clean trade partnerships.

Higher scrap charges (e.g. 75–90 %) can be leveraged in high-cost industrial regions to lower costs and global differences. However, this option is ultimately limited in total production volume in the coming decade due to limitations in scrap availability and quality, highlighting the role of additional measures, such as improved circularity and material efficiency for a Paris-compatible steel industry.

Locational cost differences also arise in labour costs and Weighted Average Cost of Capital (WACC). We estimate a global range of labour costs of €1.58 $t_{CS}^{-1}$ in India to €50.72 $t_{CS}^{-1}$ in the U.S. While we assumed a WACC of 8 % for all countries, a WACC of 5 % would lower annualised capital expenditure (CAPEX) costs for $H_2$-DRI-EAF investments by only €14.92 $t_{CS}^{-1}$. These locational differences are unlikely to drive industrial relocation on their own, whereas costs related to infrastructure such as railways, ports, pipelines, and grids, additionally increase CAPEX costs substantially if not available[61]. Regions with surplus renewable electricity through available grid connections are favourable locations for near-term deployment of $H_2$-DRI-EAF capacity and additional key national feasibility considerations include domestic political stability necessary for long-term investments. While electricity prices are the main cost component of $H_2$-DRI-EAF steel, the total power required to decarbonise existing steel assets is a comparably significant barrier for decarbonisation. Future research could therefore explore how the strength of the renewables pull correlates to interrelations of electricity prices and the volume of steel to be decarbonised, population density, and electricity consumption per capita.

Natural gas is often proposed as a transitional solution until green hydrogen becomes competitive. However, we find that NG-DRI-EAF is not competitive in the EU because of the associated EU ETS costs. Additionally, EU ETS costs would likely push NG-DRI-EAF toward NG-DRI-EAF-CCS shortly after deployment, creating new fossil fuel lock-ins and energy dependencies without zero emission potential because of partial capture rates and upstream fugitive methane emissions[8–10]. It is essential to align carbon pricing instruments with global warming potential (GWP) to avoid distorting the competitiveness between production routes simply because they emit different greenhouse gases. The EU ETS should therefore be extended to cover upstream and midstream emissions of methane, and the CBAM extended to include natural gas and methane. An extension to methane should be based on $GWP_{20}$ to reflect emission pathways consistent with the Paris Agreement and timeline to net-zero by 2050.

Natural gas-based steel exports to the EU could avoid significant CBAM costs until 2030, unless methane emissions are covered by the CBAM, and could thus become a viable option primarily for Russia, the U.S., and Turkey (see Supplementary Figs. 1, 2). While China, India, and South Korea could have competitive NG-DRI-EAF steel exports under the CBAM if scope 2 emissions are excluded and the EU ETS price is high, investments in NG-DRI-EAF do not achieve domestic competitiveness without significant carbon prices. Only exports from Russia

remain competitive if the CBAM is extended to include methane and scope 2 emissions but ultimately limited as per imposed sanctions. Moreover, NG-DRI-EAF steel exports to the EU are generally outcompeted by global $H_2$-DRI-EAF availability due to lower CBAM costs and falling costs of renewables. Corresponding investments in NG-DRI-EAF should therefore only be made with respect to domestic competitiveness. However, a detailed assessment of how natural gas could serve as such a bridge to green hydrogen, considering which combinations of LNG import infrastructure, NG-$H_2$ blends, and CCS capture rates would allow for competitive exports with the CBAM while sufficiently reducing emissions to meet climate targets, is beyond the scope of this paper.

While we model production costs and trade at country level, emission factors and potential for affordable renewable electricity vary across producers within the same country, potentially affecting intra-country competitiveness and domestic relocation for exports under the CBAM. The CBAM incentivises EU trade partners to implement strategies for retaining competitive export volumes. Most notably, countries can invest in emission-reducing technologies for single installations[62] targeted at exports, and thereby retain export volumes while most of the installed steel capacity remains emission-intensive, although product-specific production and demand may limit such straightforward substitution. Hence, carbon pricing instruments are most effective at the source of emissions i.e. the installations. Moreover, the institutional capacity and financing required for such investments render this part of the solution space non-feasible in the short term for many developing countries, especially LDCs. Thus, the CBAM is likely to have a larger impact on the export volumes of minor steelmaking and developing countries, particularly those whose exports to the EU constitute a high share of the total iron and steel export volumes[31,32]. However, many steel producing developing countries do currently not produce the specific products under the current CBAM scope[63]. Based on the revenue generated by the CBAM certificates surrendered, the EU could establish an international revenue recycling scheme for financing decarbonisation efforts[31,32,64] through e.g. bilateral programmes. Such a recycling scheme would likely increase the geopolitical acceptance of the CBAM[65]; however, require careful implementation and governance to ensure that the revenue recycled is not effectively used as a subsidy for the original source of emissions, which would undermine the CBAM altogether.

Additionally, as the CBAM allows for competitive neutrality by deducting carbon prices paid in the country of origin[20], EU trade partners can establish carbon pricing instruments to retain the revenue of the carbon cost. Countries with high trade dependency and thus vulnerability to the CBAM, would benefit the most from enhancing trade competitiveness with the EU by adopting carbon pricing instruments. However, political feasibility aspects favour countries with a synergy between high trade dependence and climate ambition, for example, Turkey[66] and South Korea[67], and are therefore likely more politically inclined to adopt such measures. Furthermore, the CBAM-supplied EU Guidance and Monitoring, Reporting, and Verification (MRV) can be adopted by countries outside of the EU for sector-specific quantification of greenhouse gas emissions, which can provide a template for establishing carbon pricing instruments and inform policymakers on aligning emission pathways with the Paris Agreement. Similarly, third countries can introduce their own CBAMs. While global differences in climate ambition and carbon prices may cause competitive disadvantages for certain countries without CBAMs, global variation in CBAM scopes and MRVs may cause substantial administrative burden and possibly restrict global trade. Instead, countries with similar climate ambition and carbon prices could establish Common Carbon Border Adjustment Mechanisms (CCBAMs), as a globally uniform carbon price would be low and thus ineffective in abating emissions from energy-intensive sectors. In conclusion, the high European demand for low-emission iron and steel may spark a decarbonisation

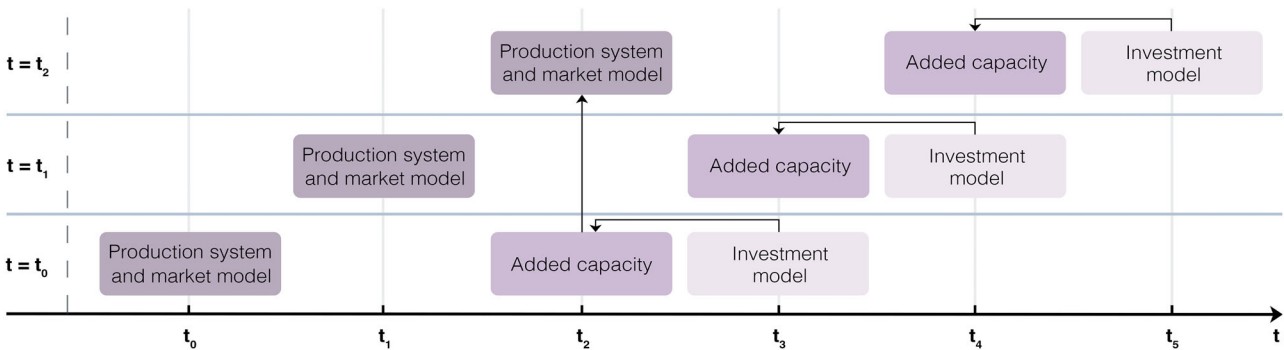

**Fig. 10 | Illustrative time-sequential model dynamics.** Example where the production system and market model gain access to the added capacity at $t = t_2$ solved for by the investment model at $t = t_3$. In the model, the investment model adds capacity in the same year it is solving for.

race among exporters, involving both technology and policy, to remain below the margin caused by EU ETS and CBAM carbon costs.

While the CBAM aims to price emissions equivalently to the EU ETS, the CBAM and the EU ETS are inherently different instruments. For example, the CBAM is based on products, whereas the EU ETS and free allocation is based on processes. Since the same product can be produced using different processes, the CBAM cost should be based on the process used and adjusted with the corresponding EU ETS benchmark to reflect free allocation in the EU. Temporary distortions in pricing can also arise from the fact that free allocation is a total amount of EUAs determined ex ante based on historical production volume (Historical Activity Level) whereas the CBAM cost is determined ex post. Since free allocation is only adjusted if the actual production volume (Actual Activity Level) deviates by ± 15%, EU producers can avoid EU ETS costs until year 2029 by producing less, whereas CBAM costs still apply. Therefore, the Actual Activity Level update threshold should be reduced or the Historical Activity Level be updated more frequently.

The overarching trade-off with the implementation of the CBAM is between comprehensiveness to minimise circumvention and competitive disadvantage for the European industry, and administrative burden by comprehensive coverage and thorough MRV. Generally, the net competitive shift is determined by the difference in coverage between the EU ETS and the CBAM, i.e. if the EU ETS applies to certain products and input materials which the CBAM does not. As different production routes use different amounts of e.g. lime, iron ore, scrap, iron, electricity, or hydrogen, any unequal pricing between the two instruments will distort the cost-competitiveness between different sub-sectors and production routes. For example, scope 2 emissions originating from electricity production are not included in the CBAM for iron, steel, and hydrogen due to World Trade Organization (WTO) compatibility and the conflict with the current European indirect cost compensation (ICC) scheme for electricity consumption[29]. As electrification as a climate mitigation strategy shifts the current scope 1 emissions from combustion to scope 2, excluding scope 2 emissions in the CBAM means surrendering certain regulatory control of the industry transition. Specifically, there is a risk of emission-intensive imports, such as HBI and $H_2$-DRI-EAF steel, where hydrogen is produced by fossil electricity. As ICC only partially covers the marginal electricity price paid by cause of ETS-induced carbon costs within the EU (15 out of 27 Member States[68]), scope 2 emissions can in theory be partially included in the CBAM whilst respecting WTO principles. However, as the current ICC is a subsidy for fossil electricity, ICC should only compensate renewable electricity or be phased out and scope 2 emissions fully included in the CBAM.

Additionally, the challenge of verification and detection of inaccurate reporting of emissions or circumvention through, e.g., recirculation of carbon prices paid in the country of origin, may also affect the net competitiveness between producers in the EU and producers in third countries. The implementation of the CBAM will likely require policy learning to refine its scope and ensure feasibility; however, any adjustments should maintain the overall policy direction to avoid undermining decarbonisation efforts and investments in both the EU and third countries.

## Methods

In order to assess the cost-competitiveness of steel-producing countries on the global market, a dynamic cost-optimisation model was developed and solved in MATLAB to minimise the discounted sum of total annual costs of the global steel industry. The model consists of two time-sequentially coupled sub-models which operate in conjunction: an investment model which solves for future added capacity which is gained three years after the Final Investment Decision (FID) to represent construction time, and a production system and market model which operates with the added capacity obtained by the investment model (see Fig. 10).

The production system and market model is a linear programming model solved using a dual-simplex algorithm, whereas the investment model is a mixed-integer linear programming model solved using a branch-and-bound framework with dual-simplex for linear programming relaxations, supported by heuristics and Gomory cuts. The optimisation variables are: added capacity, steel production, steel trade, domestic scrap and iron ore use, and scrap and iron ore trade.

All included steelmaking technologies are assumed to be integrated, BF-BOF and $H_2$-DRI-EAF make up the primary production routes and scrap-EAF the secondary production route. Installed BF capacity carries an age structure[69], added capacity can thus replace installed capacity either due to retirement or if cost-competitiveness with installed capacity is achieved. Which technological path is chosen for investment depends on its competitiveness on the domestic and international market, enabling the opportunity for industrial relocation. The national steel demands[70] must be satisfied either through domestic production or through import, the amounts of which are determined by the cost structures of the technologies, along with transportation costs and the material availability required for steel production under the operational parameters of those technologies. A maximum utilisation rate of 85% was adopted throughout the model.

### Input data

The secondary production route scrap-EAF utilises a 100% scrap charge, while the global average of 12.5%[71] was chosen for the primary steel production routes. Projections of domestic scrap availability were based on Xylia et al.[72], Nechifor et al.[73], and the Bureau of International Recycling[74], while the maximum national iron ore production capacity was calculated using a 3% Compound Annual Growth

Rate (CAGR) on national mine production data[75], and its reserve adjusted for ore quality. Furthermore, as iron ore is an essential raw material for primary steel production, additional resolution of the underlying cost structures necessary to obtain the intermediate DR-grade iron products lumps and agglomerated pellets from the run-of-mine (ROM) was incorporated (see Supplementary Fig. 3 and Supplementary Methods 4: Iron ore processing). This includes a mass flow model based on Harvey L. D. D.[76] which yields the additional cost of increased rock mining requirement and process energy consumption which arise from the mass losses originating from ore quality difference, where Fe content was used as ore quality metric. DR requires a grade of minimum 67 % Fe content and we adopt a 70 %/30 % pellet to lump ratio for the reduction shaft[77], whereas a BF typically uses a 65 % Fe content and varying amounts of sinter, pellets and lump as intermediate products for pig iron production. The model is however constrained to crushing and screening, beneficiation, and pelletising, and the iron ore deposits and grades are approximated at national level (see Supplementary Table 22).

The BF-BOF and scrap-EAF routes were powered by grid electricity in the model, while the $H_2$-DRI-EAF route was modeled with renewable electricity to allow for the condition of green hydrogen as a material requirement. Additionally, the $H_2$-DR technology was based on the continuous DRI operation assessed by Vogl et al.[78], using a specific energy consumption (SEC) of 3.48 MWh $t_{LS}^{-1}$ and 51 kg $t_{LS}^{-1}$ hydrogen consumption produced from a 72 % efficient electrolyser requiring 274 W $t_{capacity}^{-1}$ for 100 % DRI operation, which was adjusted to 235 $W\,t_{capacity}^{-1}$ to reflect the scrap charge of 12.5 %. A metallisation rate of 94 % was adopted for the reduction shaft and a metal yield of 93 % for the EAF.

In Figs. 3, 4, the low-emission $H_2$-DRI-EAF scenario is based on the HYBRIT pilot project[79,80] but where the DRI is carburised to 1.5 % C with 0.29 MWh natural gas added post reduction, which emits 59 kg $CO_2$ $t_{CS}^{-1}$. The high-emission $H_2$-DRI-EAF scenario is based on EU average emissions for input material and DRI carburised to 1.5 % C with 0.447 MWh natural gas added during reduction, which adds 169.3 kg $CO_2$ $t_{CS}^{-1}$ to a total of 437 kg $CO_2$ $t_{CS}^{-1}$. Both scenarios include 5 kg $CO_2$ $t_{CS}^{-1}$ from oxidisation of the graphite electrodes in the EAF, which are the remaining emissions if bio-coal carburises the steel in the EAF[80] (0 % C DRI) and biomethane carburises the DRI (>0 % C DRI).

We estimate emissions of 1.13 t $CO_2$ $t_{CS}^{-1}$ for the low-emission NG-DRI-EAF scenario, consistent with the Midrex NG-process of 1.1-1.2 t $CO_2$ $t_{CS}^{-1}$[81] and the lower range of IEA's estimations of 1.0-1.4 t $CO_2$ $t_{CS}^{-1}$[13]. The high-emission NG-DRI-EAF scenario with an emission factor of 1.67 t $CO_2$ $t_{CS}^{-1}$ assumes the same process emissions as the low-emission scenario but represents an EU ETS extension to methane based on $GWP_{20}$ and upstream emissions with 3 % leakage.

## Economics

All costs throughout this chapter are presented in $€_{2022}$ and installed capacity is assumed to be depreciated. The CAPEX for brownfield and greenfield BF-BOF investments were €228 and €592 $t_{CS}^{-1}$ respectively, and €247 $t_{CS}^{-1}$ for a standalone EAF. The EAF is also included in the $H_2$-DRI-EAF CAPEX of €716 $t_{CS}^{-1}$ (100 % DRI) along with the electrolyser at €0.585 $W^{-1}$ and €308 $t_{CS}^{-1}$ for the reduction shaft[78,82] (see Supplementary Table 9). Annualised capital investment costs were calculated using a discount rate equivalent to the WACC of 8 %, with a 19 year lifetime for brownfield and greenfield BF-BOF investments, and 10.5 year lifetime for relinings. 20-year lifetimes were selected for EAF and $H_2$-DRI-EAF investments, hence the CAPEX for the electrolyser was double counted (€1.17 $W^{-1}$) to account for the predicted lifetime of 10 years (80,000 operating hours)[69,78]. The NG-DRI-EAF route reforms natural gas into syngas with a reformer CAPEX of €925.47 $kW_{H_2}^{-1}$[83] for a hydrogen energy flow of 114.16 MW to supply 2.7 MWh of natural gas per tonne of steel[84]. O&M costs were calculated as 2 % of investment costs[85] for all technologies and labour costs are accounted for separately to account

for national differentiation and technology-varying labor intensities (see Supplementary Tables 17–19).

Metallurgical coal, lime, alloys, and graphite electrodes were modeled with global market prices of €175 $t_{CS}^{-1}$, €119.79 $t_{CS}^{-1}$, €2128.67 $t_{CS}^{-1}$, and €5034.23 $t_{CS}^{-1}$ respectively, whereas scrap prices, carbon prices, industrial electricity prices, natural gas prices, and prices for renewable electricity were nationally differentiated (see Supplementary Tables 10–14). Renewable electricity prices were estimated using year 2022-2023 averages of utility-scale solar PV, onshore wind power, and offshore wind power if attainable, and subjected to an exponential decline over time to represent technology learning, scale, policies, and declining installation costs. The $LCOH_2$ was calculated according to Equation (1):

$$LCOH_2 = SEC_{H_2} \times \left( \frac{CAPEX \times \left(f^{CR} + O\,\&M[\,\%]\right)}{h} + P^{Grid} \times P^{el} \right) + C^{La}$$

(1)

where $f^{CR}$ is the capital recovery factor, $SEC_{H_2}$ is the specific electricity consumption of 46.75 MWh $t_{H_2}^{-1}$[178], $h$ is the number of operating hours per year (8000), $P^{Grid}$ is a grid fee of 30 %, $P^{el}$ is the electricity price, and $C^{La}$ is the labour cost.

Export prices of the tradeable commodities steel, scrap, and iron ore were defined as the sum of the domestic market price and cost and freight (CFR). Marine transportation was the only transportation cost considered as the cost for inland transport only has a minimal effect on the total cost (global average 3 %)[15]. A shipping cost model was constructed based on travel time data, and congestion and bunkering time from Vögele et al.[86] (see Supplementary Table 15). Furthermore, we assume a Suezmax-size bulk carrier with a load volume of 160,000 t such that the Suez Canal can be utilised, powered by very-low sulphur fuel oil (VLSFO) at a price of €612 $t^{-1}$ to accommodate for future policy measures that restrict the sulfuric content of marine fuel.

## EU ETS and CBAM

The CBAM certificates to be surrendered by EU importers is phased in over the time period 2026–2034 with a CBAM factor (Article 31(1) Regulation 2023/956[20]) and free allowances received by the European steel industry sectors and sub-sectors are phased out during the same time period with the same CBAM factor (Article 10a(1a) Directive 2003/87/EC[87]). The steel industry receives free allocation based on product benchmarks. Under the system boundaries of this study, the benchmarks considered are lime, coke, sintered ore, hydrogen, hot metal, and EAF carbon steel. Free allocation is a total amount of EUAs received by each producer given by the benchmark value multiplied with the historic production volume, Historical Activity Level (HAL), and it is that total amount that decreases with the phase-out. Therefore, producers can either choose to sustain current production levels and pay a carbon cost, or produce less and remain 100 % covered by free allocation. The model accounts for both of these options. The benchmark values for the years 2021-2025 were from Regulation 2021/447[88] and the amount of free allocation post 2025 had to be estimated as it has not yet been determined by the European Commission. As the current EU ETS Directive (2003/87/EC) does not speak generic about detailed free allocation rules post 2030, a linear extrapolation of the Directive was made to fully cover the time period in which the CBAM is phased in and free allocation is phased out. That extrapolation yields years 2031–2033 as the additional period for the sintered ore, hydrogen, hot metal, and EAF carbon steel benchmarks, as the free allocation is zero starting year 2034, and 2031–2035 for the lime and coke benchmarks as they are not subjected to the phase out of free allocation. The number of CBAM certificates needed to surrender is further adjusted by the effective carbon price paid in the third country in accordance with Article 9 Regulation 2023/956[20].

Benchmark values for the periods 2026–2030 and 2031–2035 were calculated by the application of annual reduction rates. The annual reduction rates for the current period (years 2021–2025) is minimum 0.2 % and maximum 1.6 %, which is increased to 0.3 % minimum and 2.5 % maximum when determining the benchmark values post year 2025 (Article 10a(2d) Directive 2003/87/EC[87]). The annual reduction rates are based on the expected average greenhouse gas emission intensity of the 10 % most efficient installations in 2021/2022 and 2026/2027, compared to 2007/2008, for the 2026-2030 and 2031–2035 benchmarks respectively, and subsequently multiplied with the number of years between the periods (25 for years 2031–2035) and applied to the phase 3 benchmarks[88,89].

Central to this study, however, are the updated free allocation rules and the revised benchmark definitions and system boundaries thereof, most notably the addition of DRI to the hot metal benchmark, the addition of electrolysis to the hydrogen benchmark, and the addition of pellets to the sintered ore benchmark[34]. The addition of DRI to the hot metal benchmark does, however, not affect the annual reduction rate for that benchmark (Article 10a(2e) Directive 2003/87/EC[87]). Additional methodology to determine the free allocation for the three technologies included in this study under the system boundaries in Regulation 2019/331[90] and Regulation 2024/873[34], was adopted from Guidance Documents n°1, n°2, n°7, and n°9[91–94]. Furthermore, emissions originating from international maritime shipping is covered by the EU ETS 2 if the EU is a port of call and phased in according to Article 3gb(a-c) Directive 2023/959[19]. The EU ETS price was given a linear increase from €80.32 $t_{CO_2}^{-1}$ (2022)[95] to €150 $t_{CO_2}^{-1}$ (2035) and a cap of €45 $t_{CO_2}^{-1}$ was set for EU ETS 2 (Article 30h(2) Directive 2023/959[19]).

## Data availability
A dataset of the model results is available on Figshare at https://doi.org/10.6084/m9.figshare.27895656. Model input data is available in the Supplementary Information file. Source data are provided with this paper.

## Code availability
The model code developed in this study is available on Figshare at https://doi.org/10.6084/m9.figshare.27895656.

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

## Acknowledgements
C.J. thanks S. Lechtenböhmer and F. N. G. Andersson for valuable input during the early stages of this research. C.J., M.Å. and L.J.N. gratefully acknowledge the financial support received from the Swedish Energy Agency (grant number P2021-00010). Z.L. was supported by the China Scholarship Council (grant number 202006410002).

## Author contributions
C.J. Conceptualization, Methodology, Software, Validation, Formal analysis, Data Curation, Writing—Original Draft, Writing—Review and Editing, Visualization. M.Å. Validation, Writing—Original Draft, Writing—Review and Editing, Funding acquisition, Supervision. L.J.N. Writing—Original Draft, Writing—Review and Editing, Supervision. Z.L. Methodology, Supervision.

## Funding

## Competing interests
The authors declare no competing interests.
