## [Transparent Peer Review file · Nature Communications]

Emerging green steel markets surrounding the EU Emissions Trading System and Carbon Border Adjustment Mechanism

Corresponding Author: Dr Max Åhman

Version 0:

Reviewer comments:

Reviewer #1

(Remarks to the Author)

The manuscript investigates the impacts of European Emissions Trading and the Carbon Border Adjustment Mechanism on global steel markets. To this end, a techno-economic model anticipates future developments of steel production within Europe and significant steel-producing countries. Thereby, a focus is laid on shifts in production volumes and substituting BF-BOF steel production by H2-DR-EAF processes.

The manuscript addresses an interesting topic, as the impacts of the Carbon Border Adjustment Mechanism on future global steel production have not been studied in detail. Also, all recent changes in EU legislation have been considered in this study. Still, the manuscript contains some major aspects that compromise its findings. First, the literature on comparable optimization models is not provided. Also, no model formulation is given. Thus, the novelty of this optimization approach is very difficult to evaluate. Second, some major modeling assumptions are made that impact the results of this study. These assumptions need to either be justified by the authors or adjusted appropriately. If assumptions are adjusted, new results need to be obtained before publication. Thus, the study needs to be heavily revised, and additional supporting information needs to be added before it can be published in this journal.

Major remarks

- 1) This study investigates how the H2-DR-EAF might substitute future BF-BOF production due to the introduction of the European Emissions Trading System (ETS) and the Carbon Border Adjustment Mechanism (CBAM). CCS is omitted, which seems viable from an economic perspective. However, some regions will likely use high shares of natural gas in the DR process in the upcoming years. Neglecting this option has a high impact on the findings. Consequently, it needs to be included in this analysis. Alternatively, it needs to be justified, why the NG-DR-EAF process is neglected. However, I think it is very difficult to justify neglecting this alternative.
- 2) I am missing a brief overview of literature on bottom-up techno-economic optimization models to decide on favorable investments in new technologies for low-CO₂ production within energy-intensive industries such as steel. In some of these models, the impacts of the ETS are already included. This is important to differentiate this approach from previously published models.
- 3) In the introduction, the schematics of the model are illustrated. Also, some additional information are provided in the methods section and supplementary information document. However, this is not sufficient from my perspective. A mathematical model formulation must be provided with a detailed description of the developed optimization model. I suggest adding this to the supplementary methods section within the supplementary information document.
- 4) The model consists of some significant assumptions that are not justified in enough detail. For example, there is no additional explanation of the assumption of a capacitated ramp-up of H2-DR-EAF production. As these assumptions highly impact the findings, there needs to be a comprehensive explanation of the underlying considerations. Also, some assumptions might be adjusted to improve the overall quality of this study.
- 5) I am surprised that H2-DR-EAF is a viable option for European steel production in 2030/2035, even if you are not considering potential price premiums for low-CO₂ steel. In the supplementary information document, I suggest adding an overview of the annual production costs of the different process designs with and without the impacts of ETS/CBAM for each region. This might help readers to understand the economics behind your analysis. Also, an overview of your economic assessment method needs to be provided.
- 6) It is unclear what assumptions are made regarding H₂ production and resulting costs. Especially in countries with high electricity costs, importing H₂ from other regions with lower renewables costs will be favorable. Again, this significantly

impacts the development of steel production volumes and technologies in the areas you included in your study.

7) The optimization model is classified as a mixed-integer linear programming model. This model is solved using the simplex algorithm in MATLAB. Usually, the simplex algorithm can only be used to solve linear programs. Mixed-integer linear programming models are, in most cases, solved with Branch-and-Cut algorithms. However, I did not work with the optimization package in MATLAB myself. Does it allow solving mixed-integer linear programming models in a certain way? If not, changes need to be made to the manuscript.

8) You describe the optimization model as a model that consists of two parts: An investment model and a production system and market model. Is it still a single optimization model, or are two individual models solved? This needs to be clarified.

Minor remarks

1) Introduction: CCS might be an option for BF's and DR plants if natural gas is used within the process. However, as mentioned before, in both cases, CCS will most likely not be a viable option from an economic perspective.

2) Results: Your argument might be accurate that higher scrap rates are used for H2-DR-EAF steel production in the early commercialization phase. However, it needs to be clarified that the allocation under the EAF benchmarks is way lower compared to allocations that are received if sponge iron is produced.

3) Methods: The optimization model is described as a "cost-optimization model". Do you minimize costs or cash flows? This needs to be clarified if discounted cash flows are used (which would be appropriate for long-term decision-making).

4) Methods: In H2-DR-EAF steel production, 100 % HBI is used instead of hot DRI. This is only a viable option if DR plants and EAFs are not operated at the exact location since using HBI requires higher electricity than hot DRI. Why was this assumption made?

(Remarks on code availability)

Reviewer #2

(Remarks to the Author)

I would like to thank the authors and reviewers for the opportunity to review this manuscript, which is well-written and insightful. The topic is highly relevant, and the arguments are well-supported by quantitative results derived from a rigorous and transparent model. Overall, I recommend the editor to consider this work for publication but would like the authors to address the following concerns:

Abstract:

- The term "H2-DR technology" may be unfamiliar to non-experts. I recommend spelling out the full term and providing brief context to clarify what this technology entails.
- The last sentence of the abstract—"Lastly, we discuss complementing policy options to enhance the Carbon Border Adjustment Mechanism's strategic value through EU-lead global climate cooperation and the possibility of sparking an international decarbonisation race."—feels somewhat vague. It would be helpful to specify what these complementing policy options are and explain how they could enhance the CBAM's strategic value more clearly. This would leave readers with a clearer takeaway.

Introduction:

- The statement, "In our analysis we therefore assume that H2-DRI-EAF is the only alternative to conventional primary steelmaking in the period 2025-2035." is reasonable to me but may be unclear to others. I suggest elaborating on why BF-CCS is not considered a viable decarbonisation option (e.g., limitations in capture rates, construction challenges, historical failures, etc.).
- Before jumping into the EU context, I expected a more global perspective in the introduction. Given that steel is a highly traded commodity, situating the EU narrative within the broader global context could make the paper more accessible.

Results:

- The first few sentences of the results section are jargon-heavy, making them difficult to follow. I recommend starting with the key takeaways and gradually introducing the technical specifics.
- "Additionally, Europe's low grid emission intensity enables cost-competitive secondary steel production, which through an estimated increased scrap availability achieves a production increase from 54 Mt year 2025 to 68 Mt year 2035 (see Fig. 3)." While this is insightful, I wonder about the global implications of the increased EU scrap usage. How might this affect global scrap availability and trade patterns? I assume the model has some insights in this space but they are not clear in the current text.
- I would also like more details on the origin of these imports—"However, as the ramp-up of H2-DRI-EAF capacity is limited (model assumption of max 8 Mtpa, see Tab. S7), an increasing amount of imported steel will be necessary to satisfy the European steel demand (see Fig. 3 and 5a)."
- Regarding Figure 7, the terms "Demand" and "Supply" could be interpreted in various ways. I suggest defining them clearly in the figure caption to avoid ambiguity.
- Additionally, in Figure 7, it is unclear why Japan and Australia are shown without any EAF capacity in 2025. According to Table S4, both countries appear to have some EAF capacity. Could the authors clarify or correct this discrepancy?

Discussion:

- The country-specific implications presented around Figure 7 are highly engaging and deserve more attention. I recommend highlighting these implications further in the discussion section, along with clear recommendations tailored to each country.

Expanding on this could broaden the paper's appeal to a global audience.

- The authors briefly touch on steel quality, but the concept remains unclear. A more detailed explanation of what "quality" entails and why it is significant would help readers better understand this point.
- "The developing countries most notably favoured for co-location are Mauritania, Namibia, Oman, Morocco, and South Africa." This sentence feels abrupt. It would help to reference supporting information where readers can find further details.

Methods:

- The model is generally rigorous and transparent, which is great. However, the reliance on various literature sources for scrap availability and future steel demand may compromise mass-balance consistency. To improve this aspect, I recommend aligning the model with the fundamental law of mass conservation.

(Remarks on code availability)

Reviewer #3

(Remarks to the Author)

The manuscript "Emerging green steel markets surrounding the EU Emissions Trading System and Carbon Border Adjustment Mechanism" by C. Johnson, M. Åhman, L. Nilsson, and Z. Li uses a techno-economic model to study investments, production, and trade of carbon-intensive and low-carbon steel. The results provide insights into the low-carbon transition of the steel sector in Europe and trade partners. Key conclusions of the manuscript are that current EU policies (in particular regarding the allocation of free allowances and the design of the CBAM) create a window of competitiveness for hydrogen-based low-emission steel produced in the EU. Moreover, the manuscript concludes that, as this window ends due to the phasing-out of free allocations, investments into hydrogen-based low-carbon steel shifts to non-EU countries with cheap access to relevant resources (renewable energy and iron ore).

The manuscript addresses a timely and relevant topic. The methods and analysis seem mostly sound. However, some aspects of the analysis and the presentation of results should be refined before publication.

I therefore recommend publication after authors have suitably revised their manuscript.

Main points:

1) I see the main contribution of the manuscript in analysing cost competitiveness to explain investment decisions and transformation dynamics of various production routes and locations. However, the results presented mostly focus on production volumes and not on cost (only Fig. 2 provides some insight into cost). I would like to see more focus on the factors that drive the results (energy prices, input cost, carbon cost, free allocation subsidies). The results derived so far are interesting, but they seem to fall out of thin air. The manuscript pays little attention to the underlying assumptions, parameters, and drivers. For instance, I would find it helpful to see stacked-bar plots showing production cost, indicating variations between different routes, regions, and cases. In particular, I would like to know how much the free allowances subsidise production cost in comparison to other locational factors accounted for by the model. Instead, Fig. 1 could be moved to the Methods section (the usual organisation of articles published in Nature Communications is to have Methods at the end, which the manuscript otherwise follows well).

2) The manuscript rests its conclusions strongly on analysis of EU regulation. However, that regulation is only very briefly mentioned in the main text and explained in a little more detail in Methods and Supplementary Information (which I had trouble understanding). I wonder if another key contribution of the manuscript could be to explain how EU regulation (e.g. the free allowances, or the inclusion of scope 1/2 emissions into the CBAM) affect the competitiveness of different steelmaking routes in the EU and abroad. For instance, I would find it interesting to see another illustrative figure that conceptually explains the different contributions and/or a figure that demonstrates the development of these factors (various carbon costs, benefits, CBAM factors) over time. This would help the reader gain more insights about how these regulations impact cost competitiveness.

(Regarding points 1) and 2) from above, I generally suggest that the authors take a slightly more pedagogical approach and try to explain more what the underlying mechanics and drivers are of the results that they present. I found the article really very brief (which is of course great, as it is always good to be concise), but I would have appreciated a bit more elaboration of those points I mentioned above. Also, I believe that Nature Communications allows for up to 5,000 words, unlike Nature Energy or Nature Climate Change, which allow for only up to 3,000 words. This might give the authors a bit of additional space to incorporate those points.)

3) The manuscript focusses its analysis on the unabated BF-BOF route, the H₂-DRI-EAF route, and the scrap-EAF route, while noting that the BF-BOF-CCS route is ignored due to low emission-abatement potential. One key route missing from this manuscript is DRI-EAF with natural gas. I strongly suggest that the authors add it to the analysis — or that they argue clearly why it can be omitted. In particular, I am aware of multiple European steelmaking companies that plan to construct direct-reduction furnaces and initially operate them either partly or fully with natural gas. Moreover, many plants under construction will rely on hydrogen provided via a European pipeline network. Parts of the EU's hydrogen network will still be under construction and/or some direct-reduction furnaces will not have been connected to the grid by 2030, which might necessitate a bridge based on natural gas.

4) The authors briefly mention the role of shifting production patterns within the EU instead of globally (e.g. to Scandinavia or to the Iberian Peninsula), but they do not analyse it further. My intuition is that quite some of the later (beyond 2030) low-carbon investment in the European steel industry will remain within the EU but will be at different locations than today's production sites. While I appreciate the additional complexity added by an attempt to incorporate this into the existing model, I find that the absence of it presents a major deficiency for the results. I wonder if the authors could either incorporate a strongly simplified aspect of this in their model (perhaps distinguishing EU production locations into two categories: existing ones with high energy prices and potential emerging ones with low energy prices) or if the authors could at least note and underline this deficiency of their results in the main text.

5) The manuscript briefly mentions the option of importing DRI instead of steel, but this does not seem to be taken into account in the modelling. My understanding is that many existing European steelmakers are either carefully considering or actively implementing this option. I find it surprising that the presented model does not seem to incorporate this at all. Is there a way for the authors to extend their model in this regard? If this is not possible at all, could they at least address this in more detail in the main text and give an indication of how results would change if DRI trade were to be included in the model?

Other points:

- In the abstract, the manuscript claims that the European steel industry will lose competitiveness *if* the deployment of low-carbon steelmaking is too slow. This conditionality cannot be concluded clearly from the scenario design. To demonstrate this finding, I would expect to see two different scenarios, where one shows direct investment into the H2-DRI-EAF route in the period 2025–2030 due to cost competitiveness, whereas the other shows a lack of investment (e.g. due to uncertainty and lack of investment security) and a higher share of imports as a result. If the authors prefer not to introduce such a new scenario design, the statement in the abstract should be modified accordingly.
- There are two more production routes worth mentioning (but not studying in detail): a) direct electrification of primary steelmaking through molten-oxide electrolysis or alkaline electrolysis, and b) through conventional blast-furnace routes based on biomass. Route a) has the potential to be more efficient than the H2-DRI-EAF route in the long run, even though it faces low technological readiness today. Route b) is a route that is pursued strategically and is in operation today (despite concerns about direct and indirect land-use change emissions and natural habitat loss due to excessive biomass usage) in Brazil, which is one country specifically mentioned in the manuscript.
- I would like to know what differences in electricity prices and interest rates (WACC) between countries were assumed in the modelling. This information should be presented more directly in the main text.
- Fig. 4 — a) There is no gap between the two panels (top and bottom). Consequently, it looks like the axis of the bottom panel is continued in the top panel (which I don't believe is meant to be the case). This impression is further amplified by the fact that the labels and ticks on the top panel y axis are on the right-hand side. I suggest adding a gap in between the two panels and moving both axes on the left-hand side. Moreover, I think Nature's style guide has a requirement that every panel needs a label and separate description text in the figure caption. b) It might make sense to show stacked bars also in the top panel and show the scope 1 and 2 components separately.
- Trade is analysed, but aspects of resilience are not mentioned at all. Attempts to increase resilience in the EU would further support a strategy seeking to friend-shore the steel industry to other European countries (see main point 4) from above).
- The authors don't mention other subsidies in the EU, such as Industrial Projects of Common European Interest (IPCEIs). The German government funded 4 of them through direct public financing. Moreover, there are CCfDs planned. Since Germany seems to be biggest front-runner regarding H2-DRI-EAF steel at the moment, this might be worth at least mentioning, if not even partly incorporating in the model.
- The limited substitutability of primary and secondary steel is not addressed. The results show a drastic change of the share of secondary steel from 2025 to 2035. Is this in line with quality requirements of steel-consuming industries and with qualities achievable from scrap with today's technologies?
- I don't understand: "a level playing field between the EU and third countries and should thus not affect the net competitiveness". Generally, this sentence is rather long and complicated.
- "model assumption of max 8 Mt/yr" → Does this mean an annual capacity addition of 8 Mt/yr? It reads as if the model was constrained to produce no more than 8 Mt in each year, which is clearly not the case.
- To what extent is the increasing import share an outcome of the limited ramp-up rate that the model is constrained to and to what extent is it due to difference in specific production cost?
- "The CBAM was fully effective" — The use of past tense makes it sound as if the authors are referring to historical developments, when really they are referring to modelling results, correct?
- "China will most likely redirect its export" — This is not modelled in the presented manuscript, correct?
- Fig. 6 — There is no scale or axis. Would it be possible to either add one or to put numbers on top of individual flows (or do both)?
- Fig. 7 — Does "production system" mean annual production or annual production capacity? Could this be made clearer in the y axis label and unit?
- "most competitive EU trading partners" — This means exporters to the EU, right?
- I'm surprised to learn that Japan and South Korea seem to (if I understood correctly) export secondary steel to Europe. Where does that come from, given electricity prices are likely much higher than in other global regions (which is also why the H2-DRI-EAF route has little chances in Japan and South Korea).
- It is generally unclear to me how electricity costs, prices, system costs, electrolysis etc were modelled or calculated. This could be elaborated slightly more in the main text. For instance, did the authors optimise the operation of electrolysers based on hourly dispatch, or did the authors account for storage of electricity and/or hydrogen?

- What is meant by "surpass the MAC for H2-DRI-EAF investment without any or significant carbon prices due to their potential for affordable renewable electricity"?
- "considerably increase their H2-DRI-EAF capacity and countries with potential for affordable renewable electricity was identified, indicating a renewables pull effect" — This sounds as if countries with affordable renewable electricity will export either electricity or hydrogen to other countries that increase their H2-DRI-EAF capacity. But probably this is not what is meant here (and would not align with the common definition of renewables pull), right?
- "Korea secondary production is favourable through low grid electricity prices" — Could the authors clarify why this is favourable, when H2-DRI-EAF is not (as both ultimately rely on electricity)?
- "Model results highlighted China, India, the U.S., Brazil, and Australia as cost-competitive candidates in addition to Chile and Canada" → a) This statement only refers to the cited literature and not the modelling results of the presented manuscript, correct? b) I do not understand what is meant by "in addition to Chile and Canada".
- "international recycling scheme" — This is referring to recycling finance rather than steel/scrap, correct?
- I don't understand: "remain below the margin caused by CBAM carbon costs"
- "emission-intensive imports are risked by" — It sounds like there is a risk *to* emissions-intensive imports, whereas the authors probably mean that there is a risk *due to* emission-intensive imports, correct?
- There are a few (but not many) sentences in the manuscript that are rather long and therefore hard to understand. Perhaps these could be made shorter.
- I would appreciate it if the code could be made available publicly. (I would find it even more helpful if the model was created using free, open-source software, since MATLAB is a proprietary software that at least I don't have access to straight away — though I understand that this is beyond the scope of this paper.)
- Could the authors specify more clearly where they are taking the numbers for brownfield/greenfield CAPEX from?
- I don't understand: "The price for green hydrogen was modeled as a function of the renewable electricity price with a factor of 0.04675 MWh/kg".
- I don't understand: "does not speak generic about post 2030"
- Attention should be paid to where the authors refer to the EU specifically or to European countries more broadly.

Minor points and typos:

- "optimum investments" → optimal investments?
- "EU(27)" — I think "EU27" (without brackets) is a simpler and more common abbreviation.
- "Equal scrap charge ... was assumed." Either "was" → "were" (plural). Or improve sentence through use of active voice.
- Since the unit "Mtpa" is not an SI unit, I would either suggest writing "Mt/a" or "Mt/yr" or "Mt p.a".
- "12,5%" → "12.5%"
- "Bn EUR" — The common abbreviations for billion are either "B", "bil", or "bn", but I don't think I've seen "Bn" before.
- There are a few non-defining relative clauses that, I believe, require a comma:
 - "The region is however scrap-scarce which limits ..."
 - "a CBAM complemented with climate trade and diplomacy could enable critical investments which break ..."
 - "steel where hydrogen is produced"
- There are a few sentences that, I believe, need no comma at all. In particular, a serial comma is normally only used in English when writing out three or more terms.
 - "however require careful implementation, and governance, to ensure" — Probably needs no commas at all.
 - "Countries with high trade dependency, and thus vulnerable to the CBAM, would " — Probably needs no comma. Or a comma could be put around "thus". And perhaps use a substantivisation of the adjective to make the sentence easier to read: "high trade dependency and thus vulnerability".
 - "for the European industry, and administrative"
 - "through e.g. recirculation of carbon prices paid in the country of origin, may" — Either add comma before "e.g." or remove before "may".

Final remark: perhaps the authors could turn on line numbers in LaTeX, as this would make it easier for reviewers to refer to specific points in the text.

The authors are welcome to reach out to me directly, in case they have any questions or would like to discuss any of the above points directly.

Kind regards,
P.C. Verpoort

(Remarks on code availability)

Version 1:

Reviewer comments:

Reviewer #1

(Remarks to the Author)

Many thanks for revising the manuscript as well as the underlying assumptions and calculations. From my perspective, most of the reviewers' comments are addressed appropriately. However, before publication of this manuscript, some additional remarks need to be considered. The most essential comments refer to the description of your mathematical model formulation in the supplementary information.

Remarks

- 1) Introduction, lines 59–60, and Response to Reviewer comments, reviewer #1, comment 2: Thank you for including some additional studies. First of all, with my comment, I was referring to some general bottom-up techno-economic optimization models for H₂-DRI-EAF, such as in "Bhaskar, A., Abhishek, R., Assadi, M., & Somehesaraei, H. N. (2022). Decarbonizing primary steel production: Techno-economic assessment of a hydrogen based green steel production plant in Norway. *Journal of Cleaner Production*, 350. <https://doi.org/10.1016/j.jclepro.2022.131339>". These models already provide an excellent overview of the expected costs of certain production technologies. Second, in some cases, EU-ETS mechanisms are included in these techno-economic optimization models, such as in "Graupner, Y., Weckenborg, C., & Spengler, T. S. (2024). Effects of European emissions trading on the transformation of primary steelmaking: Assessment of economic and climate impacts in a case study from Germany. *Journal of Industrial Ecology*, 28(6), 1524–1540. <https://doi.org/10.1111/jiec.13544>". However, I agree with you that, to the best of my knowledge, the CBAM has not been included in any of these models. Still, referring to some of these studies in your manuscript, especially those with a strong focus on optimization, might help in pointing out the novelty of your analysis and findings.
- 2) Results, Fig. 3: Please change "Emissions" to "emissions" in the title of Subplot c.
- 3) Results, line 127: I think "from 2025 to 2026" must be replaced by "from 2025 to 2035". Also, please check Fig. 4: Why do the subplots for H₂-DRI-EAF begin with 2025, whilst those of NG-DRI-EAF start with 2026?
- 4) Supplementary information, Mathematical formulation of the model: In general, it helps the reader a lot if sets, parameters, and variables are introduced in tables before using them in the mathematical model formulation. Additionally, in this case, you do not need to explain them between your objectives and constraints; instead, you can move their description to those tables. Also, I would highly recommend structuring this section in the following way: 1.) Notation of sets, parameters, and variables, 2.) Objective function, 3.) Constraints. Currently, you are already describing some constraints before all parts of your objective function are explained. If you prefer, a separation between the descriptions of the two "sub-models" can be maintained to enhance readability.
- 5) Supplementary information, Mathematical formulation of the model: I still do not think that all economic terms are used consistently and in a correct way (see also comment 11 in the first revision). E.g., "Each sub-model minimises the net present value (NPV) of the costs of ..." on page 13: An NPV is generally calculated based on cash flows and not based on costs and revenues. Such descriptions require revision and clarification before publication. Please review the entire section on the appropriate use of the terms 'costs' and 'cash flows'. Additionally, ensure that you do not include 'investment costs' but 'investments' in the calculation of an NPV if minimization of the NPV is to be your objective function in the optimization model. Possibly, some adjustments to the objective function need to be made to consistently use either discounted cash flows as part of an NPV calculation or annual costs as the target to be minimized.

(Remarks on code availability)

Reviewer #2

(Remarks to the Author)

Thank you for the thorough revision. I believe the authors have addressed all my previous concerns.

(Remarks on code availability)

Reviewer #3

(Remarks to the Author)

The authors have addressed all concerns. I hereby recommend publication.

A few final remarks:

- The authors argue that it is fine to omit EU regions with high energy prices from their model, as this would be "equivalent". I understand the reasoning, but I would argue that it makes sense to capture such results in a model explicitly. One key purpose of a modelling exercise is to perform a combined analysis of different aspects of a complex problem. I think that there is great value in being able to say that "this option is contained in the model but not chosen in the resulting cost-optimal investment" rather than having to argue about implicit representation. This is particularly true when incorporate these options explicitly comes at little extra efforts or computational cost. Note to the editor: I don't view compliance with my suggestion a necessity for publication.
- ll. 146–147: I think there is a typo in this phrase: "emissions are lower or higher that of the EU"

Kind regards,
P.C. Verpoort

(Remarks on code availability)

Response to Reviewer comments

We thank the reviewers for their constructive and detailed comments and acknowledge the time and consideration they have devoted to evaluating our work. We have revised the manuscript in response to the feedback received and hope that we have adequately addressed their main concerns. All comments have been carefully considered and addressed, and our point-to-point response is presented below.

In response to the reviewers' comments, we have adjusted our assumptions, rerun the model, updated the results, adjusted the structure of the paper for pedagogical purposes, clarified the aim of the paper, supplemented information, and added political context in the Discussion chapter.

All differences in the manuscript compared with the previously submitted version have been highlighted in green and red to reflect added and deleted text and figures, while yellow indicates updated figures.

Reviewer	Comment #	Comment	Subject	Response
		Major remarks		
#1	1	This study investigates how the H2-DR-EAF might substitute future BF-BOF production due to the introduction of the European Emissions Trading System (ETS) and the Carbon Border Adjustment Mechanism (CBAM). CCS is omitted, which seems viable from an economic perspective. However, some regions will likely use high shares of natural gas in the DR process in the upcoming years.	Study system boundaries/assumptions	We have included techno-economic assessments of NG-DRI-EAF for all countries (see Figs. 3,4 and Supplementary Figs. 1,2). While natural gas may serve as a future transitional pathway toward green hydrogen due to resource and infrastructure constraints combined with industrial strategies and policy, we employ perfect foresight and perfect markets as optimisation models do naturally, highlighted on

	Neglecting this option has a high impact on the findings. Consequently, it needs to be included in this analysis. Alternatively, it needs to be justified, why the NG-DR-EAF process is neglected. However, I think it is very difficult to justify neglecting this alternative.	lines 280-281. Natural gas-based DRI does not become economically viable in the EU from this point of view, even under favourable natural gas and EU ETS price scenarios, unless subsidised (see Figs. 3,4). Since the cost-minimising optimisation algorithm would not invest in a technology more expensive than BF-BOF, NG-DRI-EAF can be excluded from the model because it does not influence EU results (lines 130-142). We further discuss this finding and natural gas-based opportunities for third countries for exports under the CBAM in the Discussion chapter (lines 374-385) and Supplementary Figs. 1,2. However, we wish to clarify our research question and aim, which is to assess how the EU ETS revision and CBAM impact markets and the cost-competitiveness of steel production, primarily from an EU perspective since the EU ETS and CBAM are EU policies. This is a baseline study in that regard, and we believe that we have accurately captured the main market drivers. Moreover, natural gas-based
--	---	--

			reduction will not receive free allocation from the hydrogen or synthetic gas benchmarks. Our aim is not to provide detailed decarbonisation pathways for all countries, since that would require a distinct and dedicated study. We have clarified this aim in the main text (lines 41-44). Since NG-DRI-EAF still emits 1.0-1.4 tCO₂/tSteel [1], including NG-DRI-EAF in the model would therefore require additional scenario layers of how NG-DRI-EAF could transition to near-zero emissions with analysis of combinations of LNG import supply and infrastructure, technologically feasible NG-H₂ blends, and CCS capture rates, while maintaining competitiveness with the EU ETS or CBAM. Moreover, this recent study [2] published in Nature Communications showed that steel imports are cheaper than LNG infrastructure and that “all primary crude steel [...] is imported”, which aligns with our general findings on imports. Additionally, the availability of natural gas is constrained in the EU, particularly the high-quality gas required for DR. The supply of
--	--	--	--

			natural gas is therefore currently a debated topic of energy security and resilience as a result of trade conflicts with Russia and the U.S., which together account for 35.4% of the current total supply [3]. To our knowledge, the EU does currently not have a detailed plan of how the supply of natural gas can be increased substantially to cover this potentially increased demand. Such analysis is to solve for decarbonisation pathways with energy security considerations, and thus outside the scope of this baseline study of CBAM impact. Natural gas has the highest emission reduction potential in India where it can replace coal gasification-DRI. However, since India does not have sufficient domestic supply of natural gas either, it too becomes an issue of import supply, import infrastructure, and energy security.
#1	2	I am missing a brief overview of literature on bottom-up techno-economic optimization models to decide on favorable investments in new technologies for low-CO2 production within energy-intensive industries such as steel. In some of these models, the impacts of the ETS	Lit. review A brief literature review of bottom-up techno-economic optimisation models has been added (lines 59-60). We might misunderstand which literature you are referring to but have made a substantial review and included those which are most representative and relevant for the

		are already included. This is important to differentiate this approach from previously published models.		topic of this study. In short, we find no comparable previous study that has used bottom-up modelling for assessing market perspectives under the CBAM. The only relevant bottom-up studies that we have found with the EU ETS only assume a static EU ETS price (e.g. 100 EUR/tCO₂) while our study examines the detailed EU ETS mechanisms that determine the effective EU ETS cost paid by the producer. This EU ETS legislation was adopted in May 2023 and the related Implementing and Delegated Acts and Guidance were adopted in 2024. We further wish to emphasise that we do not claim methodological novelty of the development of optimisation models for energy-intensive industries.
#1	3	In the introduction, the schematics of the model are illustrated. Also, some additional information are provided in the methods section and supplementary information document. However, this is not sufficient from my perspective. A mathematical model formulation must be provided with a detailed description of the developed	Supplementary information	We have provided a detailed mathematical formulation of the optimisation model in the Supplementary Methods section.

		optimization model. I suggest adding this to the supplementary methods section within the supplementary information document.		
#1	4	The model consists of some significant assumptions that are not justified in enough detail. For example, there is no additional explanation of the assumption of a capacitated ramp-up of H2-DR-EAF production. As these assumptions highly impact the findings, there needs to be a comprehensive explanation of the underlying considerations. Also, some assumptions might be adjusted to improve the overall quality of this study.	Assumptions	The rationale behind this constraint was to reflect real-world engineering capabilities. However, we understand the concern of this assumption having an impact on the findings. This constraint has therefore been removed for all countries except for China, since China has a national policy dictating reduction of overcapacity as well as phase-out and phase-in constraints of steel capacity. The constraint has been increased to 35 Mt p.a. which far exceeds current plans [4]. Further information and sources have been added surrounding Supplementary Tab. 7. Although this constraint has been removed for the EU, the main results remain, confirming the robustness of our results and model.
#1	5	I am surprised that H2-DR-EAF is a viable option for European steel production in 2030/2035, even if you are not considering potential price premiums for low-CO2 steel. In the supplementary information document, I suggest adding an overview of the annual production costs of the different process designs	Data	We agree that providing costs can help readers understand the economics and drivers behind the analysis and result. We have therefore considerably expanded the presented analysis and text in the manuscript in this regard. To this extent, we have added a dedicated Results section ‘Impact of EU ETS

		with and without the impacts of ETS/CBAM for each region. This might help readers to understand the economics behind your analysis. Also, an overview of your economic assessment method needs to be provided.		revision on steelmaking costs in the EU’, including additional Figs. 2,3, expanded Fig. 4 (formerly Fig. 2), added Fig. 7 to the Result section on global impact, and expanded the Methods section. The cost data for years 2026-2035 in Figs. 3,7 (line graphs) is also submitted as ‘Source Data’. We further note that it can be inferred from Fig. 2 that the free allocation subsidy for H2-DRI-EAF gained from the EU ETS revision amount to 157 EUR/tSteel (0% scrap), which is the same size as the green premium in prime EU locations such as northern Scandinavia, Portugal, and Spain (typically 100-150 EUR/tSteel), i.e. we show that the EU ETS revision replaces this previously needed green premium. Moreover, we note that this is insufficient in providing cost-competitiveness for hydrogen-based steelmaking in e.g. Germany and France where hydrogen prices are higher (lines 287-290).
#1	6	It is unclear what assumptions are made regarding H2 production and resulting costs. Especially in countries with high electricity costs, importing H2 from other regions	Assumptions	We have expanded the Methods section to explain how we calculated the LCOH2 (lines 521-526, and Equation 1). Additionally, Figs. 3,7

		with lower renewables costs will be favorable. Again, this significantly impacts the development of steel production volumes and technologies in the areas you included in your study.		now provide insights to resulting costs.
#1	7	The optimization model is classified as a mixed-integer linear programming model. This model is solved using the simplex algorithm in MATLAB. Usually, the simplex algorithm can only be used to solve linear programs. Mixed-integer linear programming models are, in most cases, solved with Branch-and-Cut algorithms. However, I did not work with the optimization package in MATLAB myself. Does it allow solving mixed-integer linear programming models in a certain way? If not, changes need to be made to the manuscript.	Clarification	We thank the reviewer for the suggestion to clarify this, which was previously concise in the interest of space. The MILP model uses a branch-and-bound framework with dual-simplex for LP relaxations supported by heuristics and Gomory cuts, whereas the LP model only uses dual-simplex. We have clarified this in the Methods section of the manuscript (lines 463-467).
#1	8	You describe the optimization model as a model that consists of two parts: An investment model and a production system and market model. Is it still a single optimization model, or are two individual models solved? This needs to be clarified.	Clarification	It is one model which consists of two sub-models. The two sub-models are coupled and cannot be operated independently of each other. Therefore, the most accurate description is that it is one model which consist of two sub-models. We agree with the value of clarifying this have improved this terminology and description in the Methods section (lines 452-460) and dedicate a figure

				to illustrate these dynamics more clearly (Fig. 10).
		Minor remarks		
#1	9	Introduction: CCS might be an option for BF's and DR plants if natural gas is used within the process. However, as mentioned before, in both cases, CCS will most likely not be a viable option from an economic perspective.	Lit. review	We agree on the economic non-viability of NG-DRI-EAF-CCS and have expanded the introduction to include this technology (lines 35-38).
#1	10	Results: Your argument might be accurate that higher scrap rates are used for H2-DR-EAF steel production in the early commercialization phase. However, it needs to be clarified that the allocation under the EAF benchmarks is way lower compared to allocations that are received if sponge iron is produced.	Data	We have adjusted our assumptions and now include EAF carbon steel for the share of scrap in H2-DRI-EAF, while noting on lines 178-180 that less free allocation will be received per tonne of steel if the scrap charge increases.
#1	11	Methods: The optimization model is described as a "cost-optimization model". Do you minimize costs or cash flows? This needs to be clarified if discounted cash flows are used (which would be appropriate for long-term decision-making).	Clarification	We optimise costs and hence use the description "cost-optimisation model", while we have further clarified this by adding that we solve for minimum global net present values (line 454).
#1	12	Methods: In H2-DR-EAF steel production, 100 % HBI is used instead of hot DRI. This is only a viable option if DR plants and EAFs are not operated at the exact location since using HBI requires higher	Assumption s	We thank the reviewer for this observation. This discrepancy has been traced to the original source who excluded the energy requirements to compact DRI to HBI. We have therefore adopted DRI

		electricity than hot DRI. Why was this assumption made?		as the system boundary for iron and adjusted the manuscript accordingly (lines 494-495).
#2	1	Abstract: The term “H2-DR technology” may be unfamiliar to non-experts. I recommend spelling out the full term and providing brief context to clarify what this technology entails.	Clarification	The wording has been changed to be more appropriate for non-experts. The sentence now reads “green hydrogen-based steelmaking” (line 11).
#2	2	Abstract: The last sentence of the abstract—“Lastly, we discuss complementing policy options to enhance the Carbon Border Adjustment Mechanism’s strategic value through EU-lead global climate cooperation and the possibility of sparking an international decarbonisation race.”—feels somewhat vague. It would be helpful to specify what these complementing policy options are and explain how they could enhance the CBAM's strategic value more clearly. This would leave readers with a clearer takeaway.	Clarification	While we agree that it is generally better to be specific, we are unfortunately limited in expanding on this by the word limit. However, we chose the wording “global climate cooperation” for this reason to begin with, as we believe the term gives a sufficient concept of the topic for non-experts.
#2	3	Introduction: The statement, "In our analysis we therefore assume that H2-DRI-EAF is the only alternative to conventional primary steelmaking in the period 2025-2035." is reasonable to me but may be unclear	Clarification	We have expanded the introduction to accommodate this and to encompass NG-DRI-EAF-CCS (lines 35-38).

		to others. I suggest elaborating on why BF-CCS is not considered a viable decarbonisation option (e.g., limitations in capture rates, construction challenges, historical failures, etc.).		
#2	4	Introduction: Before jumping into the EU context, I expected a more global perspective in the introduction. Given that steel is a highly traded commodity, situating the EU narrative within the broader global context could make the paper more accessible.	Clarification	We agree on the global context being relevant and the introduction has been expanded in this regard (lines 22-25).
#2	5	Results: The first few sentences of the results section are jargon-heavy, making them difficult to follow. I recommend starting with the key takeaways and gradually introducing the technical specifics.	Clarification	We have re-written substantial parts of the Results section with a more pedagogical approach, including the addition of a new Results section “Impact of EU ETS revision on steelmaking costs in the EU”.
#2	6	Results: "Additionally, Europe’s low grid emission intensity enables cost-competitive secondary steel production, which through an estimated increased scrap availability achieves a production increase from 54 Mt year 2025 to 68 Mt year 2035 (see Fig. 3).” While this is insightful, I wonder about the global implications of the increased EU scrap usage. How might this affect global scrap availability and trade patterns? I assume the model has	Clarification	We appreciate the reviewer’s interest in additional results of the model. We did not observe a global implication of the increased EU scrap usage because the increased use is due to an increase in EU availability, i.e. not scrap import. We have clarified that the change is due to increased domestic scrap availability (line 175).

		some insights in this space but they are not clear in the current text.		
#2	7	Results: I would also like more details on the origin of these imports —“However, as the ramp-up of H2-DRI-EAF capacity is limited (model assumption of max 8 Mtpa, see Tab. S7), an increasing amount of imported steel will be necessary to satisfy the European steel demand (see Fig. 3 and 5a).”	Assumptions	The rationale behind this constraint was to reflect real-world engineering capabilities. However, we understand the concern of this assumption having an impact on the findings. This constraint has therefore been removed for all countries except for China, since China has a national policy dictating reduction of overcapacity as well as phase-out and phase-in constraints of steel capacity. The constraint has been increased to 35 Mt p.a. which far exceeds current plans [4]. Further information and sources on this have been added before Supplementary Tab. 7.
#2	8	Results: Regarding Figure 7, the terms “Demand” and “Supply” could be interpreted in various ways. I suggest defining them clearly in the figure caption to avoid ambiguity.	Clarification	We understand the ambiguity and have therefore deleted the wording on demand as it was indeed referring to market demand rather than national steel demand. The word supply is avoided altogether (now Fig. 9).
#2	9	Results: Additionally, in Figure 7, it is unclear why Japan and Australia are shown without any EAF capacity in 2025. According to Table S4, both countries appear to have some EAF capacity. Could the authors clarify or correct this discrepancy?	Clarification	The optimal production for a given year is not necessarily all of the installed capacity. We have clarified this by adding: “The production volume correlates to the steel capacity with an 85% utilisation rate, mothballed or excess capacity is not

				displayed.” to the Figure caption of Fig. 9 (formerly Fig. 7).
#2	10	Discussion: The country-specific implications presented around Figure 7 are highly engaging and deserve more attention. I recommend highlighting these implications further in the discussion section, along with clear recommendations tailored to each country. Expanding on this could broaden the paper's appeal to a global audience.	Implications	We have expanded on recommendations for third countries, whereas fully encompassing recommendations are beyond the scope of this paper. Examples include: -U.S.: lines 242-246. -Turkey: lines 247-256. -India: lines 265-267. -Japan and South Korea: lines 277-279, 283-284.
#2	11	Discussion: The authors briefly touch on steel quality, but the concept remains unclear. A more detailed explanation of what "quality" entails and why it is significant would help readers better understand this point.	Clarification	We have tried improving on this aspect but a deep dive into the detailed concepts of scrap quality and treatment options is beyond the bounds of this paper. We have clarified the aspects we believe the reader requires to understand the key concepts and limitations: 1. Scrap quality is more clearly connected to the substitutability of primary steel (lines 33-34). 2. Scrap availability and quality is connected to circularity and material efficiency as decarbonisation pathways (lines 349-352).
#2	12	Discussion: "The developing countries most notably favoured for co-location are Mauritania, Namibia, Oman, Morocco, and South Africa.”	Clarification	We have moved this sentence to a contextually relevant paragraph and provided references for further information (line 307).

		This sentence feels abrupt. It would help to reference supporting information where readers can find further details.		
#2	13	Methods: The model is generally rigorous and transparent, which is great. However, the reliance on various literature sources for scrap availability and future steel demand may compromise mass-balance consistency. To improve this aspect, I recommend aligning the model with the fundamental law of mass conservation.	Assumptions	We agree with the reviewer that there must ultimately be mass conservation between past steel production and future scrap availability. However, since all steel products have different lifetimes and recycling efficiencies differ in all countries, it is in practice impossible to determine perfect mass-balance this way. To our knowledge, no such detailed data exists. Furthermore, we do not model past steel production (before year 2022). Additionally, future physical scrap availability, unlike future scrap use, is largely uncorrelated to future steel demand, except the market implications that more scrap decreases the scrap price and therefore may increase the demand. We have therefore ensured that the amount of future primary production and future scrap availability exceed the future steel demand. We have paid careful attention to the literature used for coherency and assumptions, and all projected future steel demand is from one source only.

#3		Major remarks		
	1	I see the main contribution of the manuscript in analysing cost competitiveness to explain investment decisions and transformation dynamics of various production routes and locations. However, the results presented mostly focus on production volumes and not on cost (only Fig. 2 provides some insight into cost). I would like to see more focus on the factors that drive the results (energy prices, input cost, carbon cost, free allocation subsidies). The results derived so far are interesting, but they seem to fall out of thin air. The manuscript pays little attention to the underlying assumptions, parameters, and drivers. For instance, I would find it helpful to see stacked-bar plots showing production cost, indicating variations between different routes, regions, and cases. In particular, I would like to know how much the free allowances subsidise production cost in comparison to other locational factors accounted for by the model. Instead, Fig. 1 could be moved to the Methods section (the usual organisation of articles published in Nature Communications is to have Methods at the end, which	Data	Please see response to comment 3. We have opted to keep Fig. 1 in the introduction as we believe it is beneficial for the reader to gain an overview of the model prior to presenting results.

		the manuscript otherwise follows well).		
	2	The manuscript rests its conclusions strongly on analysis of EU regulation. However, that regulation is only very briefly mentioned in the main text and explained in a little more detail in Methods and Supplementary Information (which I had trouble understanding). I wonder if another key contribution of the manuscript could be to explain how EU regulation (e.g. the free allowances, or the inclusion of scope 1/2 emissions into the CBAM) affect the competitiveness of different steelmaking routes in the EU and abroad. For instance, I would find it interesting to see another illustrative figure that conceptually explains the different contributions and/or a figure that demonstrates the development of these factors (various carbon costs, benefits, CBAM factors) over time. This would help the reader gain more insights about how these regulations impact cost competitiveness.	Data	Please see response to comment 3.
	3	(Regarding points 1) and 2) from above, I generally suggest that the authors take a slightly more pedagogical approach and try to explain more what the underlying	Data	We agree that providing costs can help readers understand the economics and drivers behind the analysis and result. We have therefore considerably expanded the

	mechanics and drivers are of the results that they present. I found the article really very brief (which is of course great, as it is always good to be concise), but I would have appreciated a bit more elaboration of those points I mentioned above. Also, I believe that Nature Communications allows for up to 5,000 words, unlike Nature Energy or Nature Climate Change, which allow for only up to 3,000 words. This might give the authors a bit of additional space to incorporate those points.)		presented analysis and text in the main text in this regard. To this extent, we have added a dedicated Results section ‘Impact of EU ETS revision on steelmaking costs in the EU’, including additional Figs. 2 and 3, expanded Fig. 4 (formerly Fig. 2), and added Fig. 7 to the Result section on global impact, and expanded the Methods section. Moreover, we believe that this more pedagogical approach clarifies the aim of the paper which is to analyse how the EU ETS revision and CBAM impact markets and the cost-competitiveness of steel production.
4	The manuscript focusses its analysis on the unabated BF-BOF route, the H₂-DRI-EAF route, and the scrap-EAF route, while noting that the BF-BOF-CCS route is ignored due to low emission-abatement potential. One key route missing from this manuscript is DRI-EAF with natural gas. I strongly suggest that the authors add it to the analysis — or that they argue clearly why it can be omitted. In particular, I am aware of multiple European steelmaking companies that plan to construct direct-reduction furnaces and initially operate them either partly or	Study system boundaries/assumptions	We have included techno-economic assessments of NG-DRI-EAF for all countries (see Figs. 3,4 and Supplementary Figs. 1,2). While natural gas may serve as a future transitional pathway toward green hydrogen due to resource and infrastructure constraints combined with industrial strategies and policy, we employ perfect foresight and perfect markets as optimisation models do naturally, highlighted on lines 280-281. Natural gas-based DRI does not become economically viable in the EU from this point of view, even under favourable natural

fully with natural gas. Moreover, many plants under construction will rely on hydrogen provided via a European pipeline network. Parts of the EU's hydrogen network will still be under construction and/or some direct-reduction furnaces will not have been connected to the grid by 2030, which might necessitate a bridge based on natural gas.

gas and EU ETS price scenarios, unless subsidised (see Figs. 3,4).

Since the cost-minimising optimisation algorithm would not invest in a technology more expensive than BF-BOF, NG-DRI-EAF can be excluded from the model because it does not influence EU results (lines 130-142). We further discuss this finding and natural gas-based opportunities for third countries for exports under the CBAM in the Discussion chapter (lines 374-385) and Supplementary Figs. 1,2.

However, we wish to clarify our research question and aim, which is to assess how the EU ETS revision and CBAM impact markets and the cost-competitiveness of steel production, primarily from an EU perspective since the EU ETS and CBAM are EU policies. This is a baseline study in that regard, and we believe that we have accurately captured the main market drivers. Moreover, natural gas-based reduction will not receive free allocation from the hydrogen or synthetic gas benchmarks. Our aim is not to provide detailed

			decarbonisation pathways for all countries, since that would require a distinct and dedicated study. We have clarified this aim in the main text (lines 41-44). Since NG-DRI-EAF still emits 1.0-1.4 tCO₂/tSteel [1], including NG-DRI-EAF in the model would therefore require additional scenario layers of how NG-DRI-EAF could transition to near-zero emissions with analysis of combinations of LNG import supply and infrastructure, technologically feasible NG-H₂ blends, and CCS capture rates, while maintaining competitiveness with the EU ETS or CBAM. Moreover, this recent study [2] published in Nature Communications showed that steel imports are cheaper than LNG infrastructure and that “all primary crude steel [...] is imported”, which aligns with our general findings on imports. Additionally, the availability of natural gas is constrained in the EU, particularly the high-quality gas required for DR. The supply of natural gas is therefore currently a debated topic of energy security and resilience as a result of trade conflicts with Russia and the U.S.,
--	--	--	---

			which together account for 35.4% of the current total supply [3]. To our knowledge, the EU does currently not have a detailed plan of how the supply of natural gas can be increased substantially to cover this potentially increased demand. Such analysis is to solve for decarbonisation pathways with energy security considerations, and thus outside the scope of this baseline study of CBAM impact. Natural gas has the highest emission reduction potential in India where it can replace coal gasification-DRI. However, since India does not have sufficient domestic supply of natural gas either, it too becomes an issue of import supply, import infrastructure, and energy security. Furthermore, we note that the detailed functioning of a potential hydrogen pipeline network is not yet determined and does on its own not explain where the hydrogen itself shall come from. Should the hydrogen be produced where it is competitive, i.e. northern Scandinavia, Portugal, or Spain, then that is nonetheless a form of renewables pull. Moreover, the fact
--	--	--	--

			that plants may necessitate a bridge based on natural gas is not equivalent to the production of NG-DRI-EAF being competitive, which we find it is not despite not considering infrastructure costs [2].
5	The authors briefly mention the role of shifting production patterns within the EU instead of globally (e.g. to Scandinavia or to the Iberian Peninsula), but they do not analyse it further. My intuition is that quite some of the later (beyond 2030) low-carbon investment in the European steel industry will remain within the EU but will be at different locations than today's production sites. While I appreciate the additional complexity added by an attempt to incorporate this into the existing model, I find that the absence of it presents a major deficiency for the results. I wonder if the authors could either incorporate a strongly simplified aspect of this in their model (perhaps distinguishing EU production locations into two categories: existing ones with high energy prices and potential emerging ones with low energy prices) or if the authors could at least note and underline this deficiency of their results in the main text.	Study system boundaries/assumptions	We carefully considered this option and included a similar approach when initially designing the model, where we found that approximating the EU as a single location is not a deficiency at all for this study. Since investments only occur if energy costs are sufficiently low to achieve marginal abatement, no investments occur in locations where energy costs push the steel production costs above the marginal market price. Therefore, all investments in H2-DRI-EAF occur in regions where energy costs are sufficiently low to achieve marginal abatement. This is therefore equivalent to solving for only one location which has low energy prices. Also, we found that BF-BOF production volume is invariant to the electricity price because of its low electricity consumption (lines 210-212). However, we agree with the benefit of showing how H2-DRI-EAF investments depend on hydrogen

				prices. Therefore, we vary hydrogen prices among those which enable competitive steelmaking. We now include model results for two different EU hydrogen prices, 4.12 EUR/kg and 5.5 EUR/kg derived from Fig. 4 (formerly Fig. 2) and indicate these changes in Fig. 5 (previously Fig. 3). In reality, these two hydrogen prices are found in the same locations, northern Scandinavia, Portugal, and Spain. We have generally clarified throughout the main text that this added H2-DRI-EAF capacity corresponds to these locations, which could be interpreted as the capacity shifting from existing locations with high energy prices to new locations with low energy prices as you say.
6		The manuscript briefly mentions the option of importing DRI instead of steel, but this does not seem to be taken into account in the modelling. My understanding is that many existing European steelmakers are either carefully considering or actively implementing this option. I find it surprising that the presented model does not seem to incorporate this at all. Is there are a way for the authors to extend their model in this regard? If this is not possible at all,	Study system boundaries/assumptions	We see the inclusion or exclusion of DRI-trade in the model or including it as a policy recommendation in the Discussion chapter as different methodological approaches to the same problem and underlying drivers for DRI/HBI-trade; high EU ETS costs and lacking comparative advantages for the incumbent industry. If DRI-trade is not excluded from the main scenario, it is not possible to assess the declining domestic production (see Figs. 5,6)

	could they at least address this in more detail in the main text and give an indication of how results would change if DRI trade were to be included in the model?	which occurs unless DRI-trade is actively pursued. Since no European steelmaker has a Final Investment Decision for outsourcing DRI [4] and many steelmakers strongly oppose the concept due to loss of domestic jobs (and lately also resilience), no DRI-trade is the current real-world baseline, and we therefore consider the option of DRI-trade best approached as a policy recommendation in the Discussion chapter. Along with the boundaries of our study, detailed strategies for non-competitive producers in connection to DRI has already been explored in other studies [5] and our contribution lies in assessing the underlying market drivers which in turn promote DRI-trade. Without the phase-out of free allocation there is no demand pull on the European market for low-emission production and DRI, and without the CBAM there is no business case for producing low-emission DRI outside the EU for exports to the EU market. Furthermore, since the countries with competitive H2-DRI are the countries with competitive H2-DRI-EAF, which we model, we do not
--	---	--

			believe that adding DRI-trade to the model adds significant value. We have revised the text and believe that we cover opportunities for DRI/HBI-trade in a sufficient manner. HBI-trade is introduced as a strategy on lines 281-285, elaborated on lines 290-320, and connected to resilience on lines 340-348.
		Other remarks	
7	In the abstract, the manuscript claims that the European steel industry will lose competitiveness *if* the deployment of low-carbon steelmaking is too slow. This conditionality cannot be concluded clearly from the scenario design. To demonstrate this finding, I would expect to see two different scenarios, where one shows direct investment into the H2-DRI-EAF route in the period 2025–2030 due to cost competitiveness, whereas the other shows a lack of investment (e.g. due to uncertainty and lack of investment security) and a higher share of imports as a result. If the authors prefer not to introduce such a new scenario design, the statement in the abstract should be modified accordingly.	Clarification	We believe there might be a misunderstanding of the analysed market drivers that yield our results. We have therefore added explanations in the main text for clarification (lines 144-159) and modified the statement in the abstract (lines 12-13). However, we disagree that this conditionality cannot be concluded on the basis that:  1. Our model does not demonstrate this finding, and 2. That any model is necessary to arrive at this conclusion. The provided standalone analysis of the EU ETS and fundamental economic theory is sufficient for arriving at this conclusion, given by the following explanation: Competitiveness is lost if EU

				production is more expensive than third country production. The cost of EU production is in turn dependent on the EU ETS cost, which is determined by the CBAM factor and therefore inherently time dependent. As the amount of free allocation received by the steel industry decreases every year, more and more steel production is subject to an EU ETS cost which is pushed above the marginal market price. That means that the amount of EU steel capacity which must be replaced with competitive steel capacity increases every year, given that the total installed capacity is to be retained. The production volume to be substituted is therefore time dependent and can be concluded clearly from Figs. 5 and 6 (formerly Figs. 3 and 4).
8		There are two more production routes worth mentioning (but not studying in detail): a) direct electrification of primary steelmaking through molten-oxide electrolysis or alkaline electrolysis, and b) through conventional blast-furnace routes based on biomass. Route a) has the potential to be more efficient than the H₂-DRI-EAF route in the long run, even though it faces	Lit. review	We agree that these two technologies have a significant long-term potential in decarbonisation. However: a) Given the low technology readiness of MOE and the fact that no EU steelmaker has invested in this technology [4], together with the EU ETS revision and CBAM starting next year, we see little relevance of MOE for this study. b) In the context of Brazil, it is worth

	low technological readiness today. Route b) is a route that is pursued strategically and is in operation today (despite concerns about direct and indirect land-use change emissions and natural habitat loss due to excessive biomass usage) in Brazil, which is one country specifically mentioned in the manuscript.		differentiating biochar and bio-coal for substituting coke in Blast Furnaces or producing pig iron. Thus far, only biochar has been used commercially but this has limited emission reductions (approx. 20%) compared to traditional coking coal. The emission reduction potential of bio-coal which more closely resembles the properties of coke, is significantly higher; however, is also not a direct substitute for coking coal and is currently only explored in one furnace (Tecnored) with 50-75 kt p. a. production (not a Blast Furnace). As you note, there are concerns about direct and indirect land-use change emissions. Furthermore, the sustainable outtake of biomass in Brazil is largely unquantified and the potential production volume in the short term therefore quite small. We have opted not to mention these technologies in the introduction as both require significant context with limited relevance to this study.
9	I would like to know what differences in electricity prices and interest rates (WACC) between countries were assumed in the modelling. This information should	Clarification	In the Methods section (lines 505-506) we have clarified that the discount rate used is the WACC. We now also mention this in the Discussion chapter (lines 353-364).

		be presented more directly in the main text.		
10	Fig. 4 — a) There is no gap between the two panels (top and bottom). Consequently, it looks like the axis of the bottom panel is continued in the top panel (which I don't believe is meant to be the case). This impression is further amplified by the fact the the labels and ticks on the the top panel y axis are on the right-hand side. I suggest adding a gap in between the two panels and moving both axes on the left-hand side. Moreover, I think Nature's style guide has a requirement that every panel needs a label and separate description text in the figure caption. b) It might make sense to show stacked bars also in the top panel and show the scope 1 and 2 components separately.	Clarification	a) It is indeed one panel. We show the emissions from the production and have plotted the lines above the bars for visibility. A vertical grid has been added to improve the visibility of this correlation. b) The differences between scope 1 and scope 2 emissions are too small to be visible with the order of magnitude in Fig. 5 (formerly Fig. 3). Instead, we write this difference in the text (line 205).	
11	Trade is analysed, but aspects of resilience are not mentioned at all. Attempts to increase resilience in the EU would further support a strategy seeking to friend-shore the steel industry to other European countries (see main point 4) from above).		A paragraph surrounding resilience with the example of security has been added in the Discussion chapter (lines 340-348).	
12	The authors don't mention other subsidies in the EU, such as Industrial Projects of Common European Interest (IPCEIs). The		Subsidies are now addressed on lines 330-332 and 333-339. While the German government has	

	German government funded 4 of them through direct public financing. Moreover, there are CCfDs planned. Since Germany seems to be biggest front-runner regarding H2-DRI-EAF steel at the moment, this might be worth at least mentioning, if not even partly incorporating in the model.	partially financed the construction of these projects, it is not sufficient in enabling market competitiveness for production, as evidenced by the points below. If production costs are higher than the marginal market price (i.e. negative cash flow), continuous subsidies are required as the money received otherwise runs out because there is no positive cash flow. There is no business case without positive cash flow. About the four projects mentioned: 1. ArcelorMittal has cancelled all investments in Germany despite the €1.3 billion from the German government, citing high energy and hydrogen costs [7].2. Thyssenkrupp has suspended its tender for procuring green hydrogen, citing high energy and hydrogen prices [8]. Additionally, the CEO states that they are operating beyond the limits of economic viability [9].3. Salzgitter's communicated production starting date for 100% green hydrogen-based steelmaking is 2033, a timeline sufficiently distant to raise questions surrounding the business case, since the technical lead times for electrolyzers and grid connections are significantly shorter.
--	--	---

			4. SHS's communicated production starting date is now 2028/2029 instead of 2027, without clear progress in actual construction. In contrast, Stegra (formerly H2 Green Steel) in northern Sweden is undoubtedly the front-runner in global H2-DRI-EAF developments with significant advancements in construction for production to start in 2026. Again, we believe that we have accurately captured the main market drivers with our study, that a) green steelmaking will be competitive in e.g. northern Scandinavia in 2026, and b) that green steelmaking is not competitive for hydrogen prices found in e.g. Germany. Furthermore, the EU State aid framework is currently being revised, and questions of what countries will subsidise which companies with how much for how long, is highly speculative and not sufficient for a model input. Instead, we show which hydrogen prices are necessary for a business case which can serve as basis to determine the subsidies required, although extended or major subsidies bring
--	--	--	--

				undesirable side-effects (see lines 333-339).
13	The limited substitutability of primary and secondary steel is not addressed. The results show a drastic change of the share of secondary steel from 2025 to 2035. Is this in line with quality requirements of steel-consuming industries and with qualities achievable from scrap with today's technologies?	Clarification	The limited substitutability is now addressed (lines 33-34, 349-352). While there is an increase in secondary production from 2025 to 2035, we do not characterise this as drastic, and more importantly not unreasonable given quality constraints, for the following reasons:  1. Scrap-EAF in the model also includes the long steel sector, which is 98% scrap-based already. 2. As the availability of scrap increases, so does the amount of higher quality scrap. 3. Only a minor share of current EU BF-BOF steel production is of such high quality (primarily high-strength steels AHSS, UHSS etc.) that it cannot be produced from medium to high grade scrap with a 0-10% pig iron or sponge iron input, see for example EU producer Arvedi. 4. If the circularity policies in the CID materialise, such as the ESPR, ELV Regulation revision, Circular Economy Act, etc., secondary production can increase a lot more over ten years than the 14 Mt that are 	

				our results. For example, separating wires and sensors (copper) when dismantling a car greatly improves scrap quality and therefore substitutability without additional technology requirements [10, 11]. 5. A recent study by the JRC [12] recently found a plausible increase in secondary production by 2035 similar to ours. The study notes that 50% of the total EU production volume could be scrap-based by 2035, which is roughly 63-70 Mt depending on the year of reference, whereas we found 68 Mt.
	14	I don't understand: "a level playing field between the EU and third countries and should thus not affect the net competitiveness". Generally, this sentence is rather long and complicated.	Language/Clarification	The sentence has been improved for readability (lines 144-145).
	15	"model assumption of max 8 Mtpa" → Does this mean an annual capacity addition of 8 Mt/yr? It reads as if the model was constrained to produce no more than 8 Mt in each year, which is clearly not the case.	Clarification	This sentence has been removed.
	16	To what extent is the increasing import share an outcome of the limited ramp-up rate that the model is constrained to and to what extent is it due to difference in specific production cost?	Clarification	The rationale behind this constraint was to reflect real-world engineering capabilities. However, we understand the concern of this assumption having an impact on the findings. The constraint has therefore been

				removed for all countries except for China, since China has a national policy dictating reduction of overcapacity as well as phase-out and phase-in constraints of steel capacity. The constraint has been increased to 35 Mt p.a. which far exceeds current plans [4]. Further information and sources on this have been added before Supplementary Tab. 7. Although this constraint has been removed for the EU, the results remain, confirming the robustness of our results and model.
17	"The CBAM was fully effective" — The use of past tense makes it sound as if the authors are referring to historical developments, when really they are referring to modelling results, correct?	Language		The sentence has been improved to avoid such misunderstanding (line 203).
18	"China will most likely redirect its export" — This is not modelled in the presented manuscript, correct?	Clarification		Correct, although this was a discussion misplaced in the Results section. The sentence has therefore been removed altogether.
19	Fig. 6 — There is no scale or axis. Would it be possible to either add one or to put numbers on top of individual flows (or do both)?	Clarification		A scale has been added to Fig. 8 (formerly Fig. 6).
21	Fig. 7 — Does "production system" mean annual production or annual production capacity? Could this be made clearer in the y axis label and unit?	Clarification		The word "system" has been replaced with "capacity" in Figs. 5,9 (formerly Figs. 3,7) and label of Fig. 8 (formerly Fig. 6) to avoid ambiguity.

	22 "most competitive EU trading partners" — This means exporters to the EU, right?	Clarification	The sentence has been altered so that it now specifies exporters as trade partners (lines 241-242).
	23 I'm surprised to learn that Japan and South Korea seem to (if I understood correctly) export secondary steel to Europe. Where does that come from, given electricity prices are likely much higher than in other global regions (which is also why the H2-DRI-EAF route has little chances in Japan and South Korea).	Clarification	Japan has competitive secondary steel due to low scrap prices (see Fig. 7) and South Korea has consistently been one of the largest exporters of long steel to the EU over the past decade (which is mostly secondary steel), and the largest in 2023 [6, page 41]. Overall, a lot of secondary steel will become competitive on the EU market which has previously not been competitive due to the carbon costs from the EU ETS and the CBAM, i.e. the carbon costs offset the higher electricity prices.
	24 It is generally unclear to me how electricity costs, prices, system costs, electrolysis etc were modelled or calculated. This could be elaborated slightly more in the main text. For instance, did the authors optimise the operation of electrolysers based on hourly dispatch, or did the authors account for storage of electricity and/or hydrogen?		A description and equation for how the Levelised Cost of Hydrogen was calculated has been added to the Methods section on (lines 521-526) and Equation 1.
	25 What is meant by "surpass the MAC for H2-DRI-EAF investment without any or significant carbon prices due	Clarification	The sentence has been adjusted to more clearly reflect that these countries invested in H2-DRI-EAF (surpassed the marginal abatement

		to their potential for affordable renewable electricity"?		cost) because of affordable electricity despite not being subjected to significant or any carbon costs (lines 258-260).
	26	"considerably increase their H2-DRI-EAF capacity and countries with potential for affordable renewable electricity was identified, indicating a renewables pull effect" — This sounds as if countries with affordable renewable electricity will export either electricity or hydrogen to other countries that increase their H2-DRI-EAF capacity. But probably this is not what is meant here (and would not align with the common definition of renewables pull), right?	Language/cl arification	The sentence has been corrected to more clearly state that the countries with a considerable increase in H2-DRI-EAF capacity are those that have affordable renewable electricity, along the common definition of renewables pull (lines 260-262).
	27	"Korea secondary production is favourable through low grid electricity prices" — Could the authors clarify why this is favourable, when H2-DRI-EAF is not (as both ultimately rely on electricity)?	Clarification	First, H2-DRI-EAF requires five times more electricity than a standalone EAF. Electricity prices sufficiently low for secondary production may in general therefore not be sufficiently low for competitive H2-DRI-EAF production. Second, H2-DRI-EAF investment adds a capital investment cost whereas the installed EAF for secondary production is already depreciated.
	28	"Model results highlighted China, India, the U.S., Brazil, and Australia as cost-competitive candidates in addition to Chile and Canada" → a)	Clarification	The sentence has been altered to improve readability (lines 303-305).

		This statement only refers to the cited literature and not the modelling results of the presented manuscript, correct? b) I do not understand what is meant by "in addition to Chile and Canada".		
	29	"international recycling scheme" — This is referring to recycling finance rather than steel/scrap, correct?	Clarification	Correct. The word “revenue” has been added to clarify this (line 399).
	31	I don't understand: "remain below the margin caused by CBAM carbon costs"	Clarification	An explanation of the underlying mechanisms of this market dynamic has been added (lines 144-159).
	32	"emission-intensive imports are risked by" — It sounds like there is a risk *to* emissions-intensive imports, whereas the authors probably mean that there is a risk *due to* emission-intensive imports, correct?	Language	The sentence has been re-written for clarity (lines 420-422).
	33	There are a few (but not many) sentences in the manuscript that are rather long and therefore hard to understand. Perhaps these could be made shorter.	Language	We understand it was difficult to point these out without line numbers enabled previously. Attention has been paid to the length of sentences throughout the manuscript.
	34	I would appreciate it if the code could be made available publicly. (I would find it even more helpful if the model was created using free, open-source software, since MATLAB is a proprietary software that at least I don't have access to straight away — though I understand	Clarification	The code was and remains publicly available on Figshare, previously found in section ‘S2.3 Code availability’ in Supplementary Information and now accessible through the ‘Code Availability’ section on page 21 in the manuscript.

		that this is beyond the scope of this paper.)		
35		Could the authors specify more clearly where they are taking the numbers for brownfield/greenfield CAPEX from?	Clarification	A reference to Supplementary Tab. 9 (previously Supplementary Tab. 8) has been added.
36		I don't understand: "The price for green hydrogen was modeled as a function of the renewable electricity price with a factor of 0.04675 MWh/kg".	Clarification	This sentence has been removed. Instead, the specific energy consumption of the electrolyser is more clearly explained in the paragraph below (line 524).
37		I don't understand: "does not speak generic about post 2030"	Clarification	The sentence has been clarified (line 546). In essence, the Directive does not lay out overarching rules for the functioning of the EU ETS beyond 2030, except for the CBAM factor.
38		Attention should be paid to where the authors refer to the EU specifically or to European countries more broadly.	Clarification	We differentiate between the EU and European countries on the basis that the EU ETS applies to some European countries that are not in the EU, whereas the CBAM only applies to EU Member States.
		Minor points and typos		
39		"optimum investments" → optimal investments?	Language	This suggested change has been implemented (line 71).
41		"EU(27)" — I think "EU27" (without brackets) is a simpler and more common abbreviation.	Language	The notation EU27 has been adopted throughout the manuscript.
42		"Equal scrap charge ... was assumed." Either "was" → "were" (plural). Or improve sentence through use of active voice.	Language	“was” has been changed to “were”.

43	Since the unit "Mtpa" is not an SI unit, I would either suggest writing "Mt/a" or "Mt/yr" or "Mt p.a.".	Language	The notation "Mt p. a." has been adopted throughout the manuscript.
44	"12,5%" → "12.5%"	Language	The entire sentence has been removed (line 177).
45	"Bn EUR" — The common abbreviations for billion are either "B", "bil", or "bn", but I don't think I've seen "Bn" before.	Language	The paragraph with this notation has been removed (lines 213-216).
46	There are a few non-defining relative clauses that, I believe, require a comma:  - "The region is however scrap-scarce which limits ..." - "a CBAM complemented with climate trade and diplomacy could enable critical investments which break ..." - "steel where hydrogen is produced" 	Language	Commas have been added in the suggested sentences.
47	There are a few sentences that, I believe, need no comma at all. In particular, a serial comma is normally only used in English when writing out three or more terms.  - "however require careful implementation, and governance, to ensure" — Probably needs no commas at all. - "Countries with high trade dependency, and thus vulnerable to the CBAM, would " — Probably needs no comma. Or a comma could be put around "thus". And perhaps 	Language	Selected commas have been removed from the suggested sentences.

	use a substantivisation of the adjective to make the sentence easier to read: "high trade dependency and thus vulnerability". - "for the European industry, and administrative" - "through e.g. recirculation of carbon prices paid in the country of origin, may" — Either add comma before "e.g." or remove before "may".		
48	Final remark: perhaps the authors could turn on line numbers in LaTeX, as this would make it easier for reviewers to refer to specific points in the text.	Other	Line numbers have been turned on.
49	The authors are welcome to reach out to me directly, in case they have any questions or would like to discuss any of the above points directly. Kind regards, P.C. Verpoort	Other	We appreciate being given the opportunity to reach out directly; however, have not found this necessary.

References

- [1] IEA. Iron and Steel Technology Roadmap. <https://www.iea.org/reports/iron-and-steel-technology-roadmap> (IEA, Paris, 2020).
- [2] Neumann F., Hampp J. & Brown T. Green energy and steel imports reduce Europe's net-zero infrastructure needs. *Nat. Commun.* **16**, 5302 (2025).
- [3] European Commission. *Where does the EU's gas come from?* 2025. <https://www.consilium.europa.eu/en/infographics/where-does-the-eu-s-gas-come-from/#0>
- [4] LeadIT. Green Steel Tracker 2025. <https://www.industrytransition.org/green-steel-tracker/>.
- [5] Bilici S. et al. Global trade of green iron as a game changer for a near-zero global steel industry? - A scenario-based assessment of regionalized impacts. *Egy. Clim. Change.* **5**, 100161 (2024).
- [6] European Steel Association. *European Steel in Figures 2024* 2024. <https://www.eurofer.eu/assets/publications/brochures-booklets-and-factsheets/european-steel-in-figures-2024/EUROFER-2024-Version-June14.pdf>
- [7] Fastmarkets. *European steel industry: ArcelorMittal halts EAF/DRI project* 2025. <https://www.fastmarkets.com/insights/european-steel-industry-arcelormittal-halts-eaf-dri-project/>
- [8] Hydrogen Insight. *Thyssenkrupp has put hydrogen tender for €3bn green steel plant , on hold' due to 'significantly higher' prices than expected* 2025. <https://www.hydrogeninsight.com/industrial/exclusive-thyssenkrupp-has-put-hydrogen-tender-for-3bn-green-steel-plant-on-hold-due-to-significantly-higher-prices-than-expected/2-1-1795894>
- [9] Reuters. *Thyssenkrupp can't guarantee \$3.3 bln green steel site will be economical, CEO says* 2025. <https://www.reuters.com/markets/commodities/thyssenkrupp-cant-guarantee-33-bln-green-steel-site-will-be-economical-ceo-says-2025-03-19/>
- [10] Gross H. & Hermine J.P. Car-to-Car Steel: Potential of End-of-Life Vehicle deep-dismantling and use of copper depolluted steel scrap to decarbonize automotive flat steel production. IMT-IDDR (2025). https://institut-mobilites-en-transition.org/wp-content/uploads/2025/03/IMT_Car-to-car_Steel_final.pdf
- [11] Sutter J., Adjei F., Baron Y. & Kosińska-Terrade I. Boosting the use of recycled steel in the EU automotive industry under the ELV Regulation. Oeko-Institut (2025). https://www.transportenvironment.org/uploads/files/2025_04_Report_Recycled_steel_EU_automotive_industry_final.pdf

[12] Maury T., Torres de Matos C., Blanco Perez S., Arcipowska A., Moya J. et al., Analysis of the EU Steel supply chain: current trends and circularity opportunities - Raw Material Information System Brief. (European Commission, Ispra, JRC142660, 2025).

<https://publications.jrc.ec.europa.eu/repository/handle/JRC142660>

Response to Reviewer comments

We thank the reviewers for their further efforts and feedback and acknowledge the time and consideration they have devoted to evaluating our work. We have revised the manuscript in response to the feedback received and hope that we have adequately addressed their main concerns. All comments have been carefully considered and addressed, and our point-to-point response is presented below.

In addition to the changes made according to the suggestions made by the reviewers, we have made the following three minor edits to the manuscript:

1. To improve readability and limit the word count, the description of the scenarios in Figs. 3 and 4 has been moved from the Results section to the Methods section (lines 99-104 and 111-114 moved to lines 461-471), and a reference to Fig. 4 has been included (line 461).
2. To correct a dangling modifier, the paragraph in lines 154-159 has been moved to lines 146-151 above the paragraph in lines 151-153. The wording of the paragraph in lines 151-153 has subsequently been re-worded to improve flow.
3. Minor grammatical and language edits.

All differences in the manuscript compared with the previously submitted version have been highlighted in green and red to reflect added and deleted text, while yellow indicates the updated titles of the subplots in Fig. 3. Our response to the reviewers' comments is written in blue.

Point-to-point response

Reviewer #1

Many thanks for revising the manuscript as well as the underlying assumptions and calculations. From my perspective, most of the reviewers' comments are addressed appropriately. However, before publication of this manuscript, some additional remarks need to be considered. The most essential comments refer to the description of your mathematical model formulation in the supplementary information.

We are very grateful you believe that we have addressed most of the reviewers' appropriately and for your further checking of our revised manuscript. We agree with your suggestions and have revised the manuscript and Supplementary Information accordingly.

Remarks

1) Introduction, lines 59–60, and Response to Reviewer comments, reviewer #1, comment 2: Thank you for including some additional studies. First of all, with my comment, I was referring to some general bottom-up techno-economic optimization models for H2-DRI-EAF, such as in "Bhaskar, A., Abhishek, R., Assadi, M., & Somehesaraei, H. N. (2022). Decarbonizing primary steel production: Techno-economic assessment of a hydrogen based green steel production plant in Norway. *Journal of Cleaner Production*, 350. <https://doi.org/10.1016/j.jclepro.2022.131339>". These models already provide an excellent overview of the expected costs of certain production technologies. Second, in some cases, EU-ETS mechanisms are included in these techno-economic optimization models, such as in "Graupner, Y., Weckenborg, C., & Spengler, T. S. (2024). Effects of European emissions trading on the transformation of primary steelmaking: Assessment of

economic and climate impacts in a case study from Germany. *Journal of Industrial Ecology*, 28(6), 1524–1540. <https://doi.org/10.1111/jiec.13544>". However, I agree with you that, to the best of my knowledge, the CBAM has not been included in any of these models. Still, referring to some of these studies in your manuscript, especially those with a strong focus on optimization, might help in pointing out the novelty of your analysis and findings.

Thank you for clarifying which articles you were referring to with your previous comment. We agree that referring to these studies help pointing out the novelty of our analysis and have included both mentioned studies in the Introduction under the literature review section (line 57 for Bhaskar et al. and lines 65-68 for Graupner et al.).

We however wish to note that while a variety of such country-specific studies using bottom-up techno-economic optimisation models exist, these do not account for external market conditions, e.g. that domestic competitiveness is also dependent on international competition, and thus have limited overlap with our study. We model international trade and hence consider global competition between all countries of all types of steel production included in the model, and the CBAM is an add-on to that. We believe that your comment has helped us communicate this more clearly.

2) Results, Fig. 3: Please change "Emissions" to "emissions" in the title of Subplot c.

Thank you for pointing out this typo. We have capitalised "emissions" in Fig. b,e,f, for coherence with Fig. 4.

3) Results, line 127: I think "from 2025 to 2026" must be replaced by "from 2025 to 2035". Also, please check Fig. 4: Why do the subplots for H2-DRI-EAF begin with 2025, whilst those of NG-DRI-EAF start with 2026?

a) We believe there might be a misunderstanding of what we mean by lines 120-122. We have therefore changed the phrasing to more clearly communicate that we are referring to the impact of the EU ETS revision, which enters into force in 2026. I.e. how big of a difference the EU ETS revision makes compared to no EU ETS revision (year 2025). This is mainly written to quantify and more clearly explain the differences visible in Figs. 4a,b between 2025 and 2026.

b) Due to higher emissions and no free allocation in 2025, there is a large difference in total cost between 2025 and 2026. Including 2025 therefore makes the year 2025 take up 84 % of the colourbar and hence making variations over the period 2026-2035 not clearly visible. The differences of 2025-2026 is nonetheless presented in Fig. 3, through comparison of Figs. 3e,f with Fig. 3a. We have now additionally written the 2025 values in the caption of Fig. 4. For clarification: there is no (positive) natural gas price for which the NG-DRI-EAF route obtains competitiveness in 2025.

4) Supplementary information, Mathematical formulation of the model: In general, it helps the reader a lot if sets, parameters, and variables are introduced in tables before using them in the mathematical model formulation. Additionally, in this case, you do not need to explain them between your objectives and constraints; instead, you can move their description to those tables. Also, I would highly recommend structuring this section in the following way: 1.) Notation of sets, parameters, and variables, 2.) Objective function, 3.) Constraints. Currently, you are already describing some constraints before all parts of your objective function are explained. If you prefer, a separation between the descriptions of the two "sub-models" can be

maintained to enhance readability.

Thank you for your tips on how we can improve the presentation of our mathematical formulation of the model. We have implemented all suggested changes, except for separating the two sub-models, which we kept together for pedagogical purposes to more clearly explain the links between the two. All notation is now introduced in tables (Supplementary Tabs. 24-27) and the constraints are described after the objective function. We chose to keep some descriptions of functions, variables, and sets when listing the Supplementary Equations, so the reader does not need to scroll back to the tables where they are defined.

5) Supplementary information, Mathematical formulation of the model: I still do not think that all economic terms are used consistently and in a correct way (see also comment 11 in the first revision). E.g., "Each sub-model minimises the net present value (NPV) of the costs of..." on page 13: An NPV is generally calculated based on cash flows and not based on costs and revenues. Such descriptions require revision and clarification before publication. Please review the entire section on the appropriate use of the terms 'costs' and 'cash flows'. Additionally, ensure that you do not include 'investment costs' but 'investments' in the calculation of an NPV if minimization of the NPV is to be your objective function in the optimization model. Possibly, some adjustments to the objective function need to be made to consistently use either discounted cash flows as part of an NPV calculation or annual costs as the target to be minimized.

We appreciate the opportunity to clarify the economic terms used when describing the model. We have altered the description in the Methods section (lines 421-422) and the description of the mathematical formulation in the Supplementary Information. We now consistently refer to "the minimisation of the discounted sum of total annual costs". This formulation accurately describes the objective functions as written in Supplementary Equations 9,10.

Reviewer #2

Thank you for the thorough revision. I believe the authors have addressed all my previous concerns.

We thank the reviewer for their input, and help in improving this manuscript.

Reviewer #3

The authors have addressed all concerns. I hereby recommend publication.

We thank you for your constructive and detailed comments which we believe have significantly improved this manuscript, and are grateful you recommend publication.

A few final remarks:

- The authors argue that it is fine to omit EU regions with high energy prices from their model, as this would be "equivalent". I understand the reasoning, but I would argue that it makes sense to capture such results in a model explicitly. One key purpose of a modelling exercise is to perform a combined analysis of different aspects of a complex problem. I think that there is great value in being able to say that "this option is contained in the model but not chosen in the resulting cost-optimal investment" rather than having to argue about implicit representation. This is particularly true when incorporate these options explicitly comes

at little extra efforts or computational cost. Note to the editor: I don't view compliance with my suggestion a necessity for publication.

While we agree there is a benefit of direct over implicit representation, adding regions to the model unfortunately requires substantial extra efforts not feasible at this time. We appreciate that you agree that this would not change the presented results and that you do not view compliance with your suggestion a necessity for publication.

- ll. 146–147: I think there is a typo in this phrase: "emissions are lower or higher that of the EU"

Thank you for this observation. The typo has been corrected (line 142).

Kind regards,
P.C. Verpoort